# Enhancing Cost Efficiency in Active Learning with Candidate Set Query

## Abstract

This paper introduces a cost-efficient active learning (AL) framework for classification, featuring a novel query design called *candidate set query*. Unlike traditional AL queries requiring the oracle to examine all possible classes, our method narrows down the set of candidate classes likely to include the ground-truth class, significantly reducing the search space and labeling cost. Moreover, we leverage conformal prediction to dynamically generate small yet reliable candidate sets, adapting to model enhancement over successive AL rounds. To this end, we introduce an acquisition function designed to prioritize data points that offer high information gain at lower cost. Empirical evaluations on CIFAR-10, CIFAR-100, and ImageNet64x64 demonstrate the effectiveness and scalability of our framework. Notably, it reduces labeling cost by 42% on ImageNet64x64.

## 1 Introduction

Deep neural networks owe much of their success to large-scale annotated datasets (Deng et al., 2009b; Kirillov et al., 2023; OpenAI, 2023; Radford et al., 2021). Scaling datasets is crucial for improving both of their performance (Hestness et al., 2017; Zhai et al., 2022) and robustness (Fang et al., 2022). However, the resources demanded for manual annotation pose a significant bottleneck, particularly in fields requiring expert input like medical data. In response to these challenges, cost-efficient methods for dataset collection, such as semi-automatic labeling (Kim et al., 2024; Qu et al., 2024; Wang et al., 2024), synthetic data generation (Liu et al., 2019; Tran et al., 2019), and active learning (AL) (Ash et al., 2020; Kirsch et al., 2019; Sener & Savarese, 2018; Settles, 2009; Sinha et al., 2019; Wang & Ye, 2015) have been studied.

This paper investigates AL for classification, where a training algorithm selects informative samples from the data pool and queries annotators for their class labels within a limited budget. We focus on improving the design of annotation queries, emphasizing their critical role. To be specific, we consider image classification of $L$ classes. In a conventional design of query, an annotator is asked to choose a class in the list of $L$ classes. Here, the effort needed to review the entire class list and identify the correct class increases as the list size $L$ increases; according to an information-theoretic analysis (Hu et al., 2020), the cost of choosing among $L$ options is $\log_2 L$. To address this issue of growing annotation cost, recent studies (Hu et al., 2020; Kim et al., 2024) employ a 1-bit query design asking annotators to check if the top-1 model prediction is correct. While this simplifies and speeds up annotation, it produces weak supervision incompatible with standard classification loss functions, necessitating specialized losses and algorithms like contrastive loss and semi-supervised learning techniques.

We propose *candidate set query* (CSQ), a novel AL query design that remains cost-efficient with increasing classes and integrates seamlessly with existing loss functions. CSQ presents the annotator with an image and a narrowed set of candidate classes, which is likely to include the ground-truth class. If the ground-truth class is within these candidates, the annotator selects from this smaller group; otherwise, they select from the remaining classes. This query approach can reduce labeling costs by reducing the search space required for annotation, particularly effective in scenarios with a wide range of classes where the search space for the annotator would be extensive. Fig. 1(*left*) compares CSQ with the conventional query in AL for classification to show its efficiency.

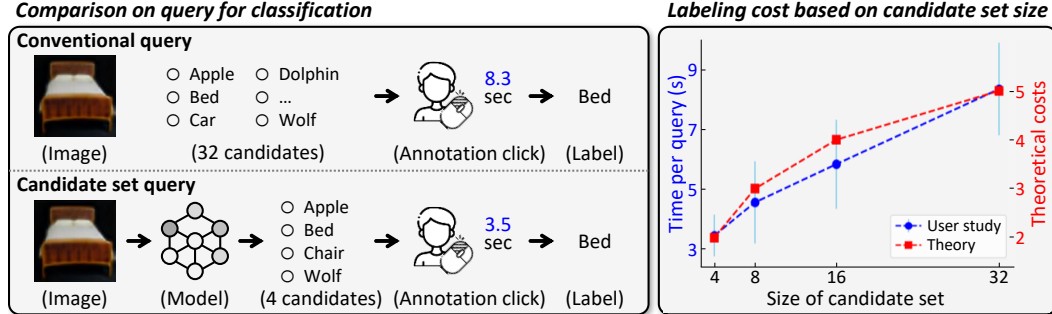

Figure 1: Conventional query versus CSQ. (*left*) While the conventional query presents all possible options to annotators, CSQ leverages the knowledge of model to offer narrowed options that are likely to include the true label, thereby reducing the annotation time. (*right*) By conducting a user study on 40 participants, we demonstrate that the labeling cost increases logarithmically to the candidate set size, which closely aligns with the information theoretic cost suggested by Hu et al. (2020) with a correlation coefficient of 0.97. Note that as the labeling cost increases per sample, the overall labeling cost increases significantly when multiplied by the total number of labeled samples. Further details of the user study are provided in Sec. 4.2 and Appendix A.

In the CSQ framework, the design of the candidate set is crucial for its effectiveness. Too many candidates unnecessarily increase the labeling costs. On the other hand, too few candidates are likely to omit the ground-truth class, requiring additional queries to identify the true class among the remaining classes, which can sometimes be more expensive than the conventional query. To enhance the effectiveness of the CSQ framework, we propose to construct candidate sets guided by prediction uncertainty from a trained model using conformal prediction (Angelopoulos et al., 2023). Conformal prediction aims at constructing a set of predictions including the true class, where each set is properly sized based on the certainty of the model about the input. This strategy enables flexible adjustment of the candidate set for each sample, expanding it for an uncertain sample to include the true label and shrinking it for more certain one to reduce the labeling cost. Furthermore, we optimize the level of certainty in conformal prediction to minimize the labeling cost for each round. Therefore, this candidate set construction adapts to the increasing accuracy of the model over successive AL rounds, refining the candidate set as the model improves.

Last but not least, we propose a new acquisition function designed to maximize the cost efficiency of CSQ. Conventional acquisition functions in AL are designed to favor samples with high estimated information gain, assuming uniform annotation costs across all samples. On the other hand, in CSQ, the labeling cost for each sample varies according to the size of its candidate set. Thus, we propose an acquisition function that evaluates samples based on the ratio of estimated information gain to labeling cost. Specifically, we combine the conventional acquisition function score, which indicates the estimated information gain, with the estimated cost derived from the candidate set, favoring samples that maximize information gain per unit cost. This cost-efficient acquisition function can incorporate with any sample-wise acquisition score, ensuring the selection of both informative and cost-efficient samples.

The proposed method achieved state-of-the-art performance on CIFAR-10 (Krizhevsky et al., 2009), CIFAR-100 (Krizhevsky et al., 2009), and ImageNet64x64 (Chrabaszcz et al., 2017). We verify the effectiveness and robustness of CSQ through extensive experiments with varying datasets, acquisition functions, and budgets. Notably, CSQ achieves the same performance as the conventional query on ImageNet64x64 at only 42% of the cost, showing its scalability. Ablation studies demonstrate that both our candidate set construction and sampling strategy contribute to the performance. Further, the necessity of CSQ is demonstrated by a user study involving 40 participants. In short, the main contribution of this paper is four-fold:

- We propose a novel query design for active learning, where the annotator is presented with an image and a narrowed set of candidate classes that are likely to include the ground-truth class. This approach, termed CSQ, significantly reduces labeling cost by minimizing the search space the annotator needs to explore.

- To maximize the advantage of CSQ, we propose to utilize conformal prediction to dynamically generate small yet reliable candidate sets optimized to reduce labeling costs, adapting to the evolving model throughout successive AL rounds.

- We propose a new acquisition function that prioritizes data points expected to have high information gain relative to their labeling costs, enhancing cost-efficiency.

- The proposed framework achieved state-of-the-art performance on diverse image recognition datasets, CIFAR-10, CIFAR-100, and ImageNet64x64, showing its effectiveness and generalizability.

## 2 RELATED WORK

**Acquisition functions in AL.** AL is a well-established problem Settles (2009); Dasgupta (2011); Hanneke et al. (2014) that focuses on selectively querying the most informative samples for annotation to maximize model performance within a limited budget. To assess informativeness, various acquisition functions have been proposed, considering either the uncertainty of model predictions (Asghar et al., 2017; He et al., 2019; Ostapuk et al., 2019; Fuchsgruber et al., 2024), diversity in feature space (Sener & Savarese, 2018; Sinha et al., 2019; Yehuda et al., 2022), or both (Ash et al., 2020; Hwang et al., 2022; Wang & Ye, 2015; Wang et al., 2019). Disagreement-based AL and its variants are supported by rigorous theoretical learning guarantees (Hanneke et al., 2014; Krishnamurthy et al., 2019). Recent studies have demonstrated that the choice of acquisition functions depends on the budget, with uncertainty being more suitable for a high budget and typicality for a low budget (Hacohen et al., 2022; Hacohen & Weinshall, 2023a). In addition, a look-ahead acquisition function that considers nearby samples simultaneously (Kim et al., 2024) and the selection of easily flip-flopped samples (Cho et al., 2024) have also been proposed. However, these methods assume that all samples require the same cost and select samples based solely on the amount of information. We point out that the cost required for each sample can vary and prioritize selecting samples that offer the most information considering their cost.

**Conformal prediction (CP).** CP enables us to quantify uncertainty in predictions with associated confidence levels (Shafer & Vovk, 2008). Recent advances in CP empower classifiers to generate predictive sets that include the true label with a probability chosen by the user (Angelopoulos et al., 2020). Additionally, in the field of AL, nonconformity measurements from CP are employed in the acquisition function to select informative samples (Matiz & Barner, 2020). In contrast, we utilize CP not only to develop a cost-efficient acquisition function but also to design an efficient candidate set query reducing the labeling cost.

**Efficient query design.** Designing efficient annotation queries reduces the annotation costs of crafting datasets. In various computer vision tasks, diverse types of queries have been investigated, including conventional classification queries (Hacohen & Weinshall, 2023b) requiring a specific class, one-bit queries (Hu et al., 2020) asking for yes or no answers, multi-class queries (Hwang et al., 2023) identifying all classes within a set of multiple instances, and correction queries (Kim et al., 2024) utilizing pseudo labels from the model. However, existing queries remain stagnant in their predefined forms regardless of the model's performance improvement in successive AL rounds. The proposed candidate set query is cost-efficient while provides complete supervision which can be integrated seamlessly with existing loss functions.

## 3 PROPOSED METHOD

We consider general classification tasks such that for input $\mathbf{x}$ and a categorical variable $y \in \mathcal{Y} = \{1, 2, \ldots, L\}$, a model parameterized by $\boldsymbol{\theta}$ predicts the class of the input as $\arg\max_{y \in \mathcal{Y}} P_{\boldsymbol{\theta}}(y|\mathbf{x})$. We study an active learning (AL) scenario conducted over $R$ rounds. In each round $r$, a budget of $B$ samples is actively selected from the unlabeled data pool $\mathcal{X}$ using an acquisition function. This actively selected set $\mathcal{A}_r$ is then labeled by an annotator to form the labeled dataset $\mathcal{D}_r$ with labeling cost $C_r$, and is used to update the model. Let $\boldsymbol{\theta}_r$ denote the model trained on the accumulated labeled data up to round $r$, $\bigcup_{i=0}^{r} \mathcal{D}_i$. Our goal is to maximize the performance of $\boldsymbol{\theta}_r$, while minimizing the accumulated cost $\bigcup_{i=0}^{r} C_i$. The key aspect of the proposed method is the candidate set query (CSQ), which reduces $C_r$ by narrowing the set of candidate classes presented to annotators. For simplicity, we omit the round index $r$ from $\boldsymbol{\theta}_r$ in the remainder of this section.

---

**Algorithm 1** Active learning with candidate set query

---

**Require:** The number of active learning rounds $R$, round-wise budget $B$, unlabeled data pool $\mathcal{X}$, randomly sampled initial labeled dataset $\mathcal{D}_0$.
1: Train the initial model $\boldsymbol{\theta}_0$ on $\mathcal{D}_0$.
2: **for** $r = 1, 2, \ldots, R$ **do**
3:     Select the top $B$ samples $\mathcal{A}_r \subset \mathcal{X}$ with highest acquisition scores $g_{\text{cost}}(\mathbf{x})$.     ▷ Sec. 3.3
4:     Construct cost-efficient candidate set $\hat{Y}(\mathbf{x})$ for each $\mathbf{x} \in \mathcal{A}_r$.     ▷ Sec. 3.2
5:     Query annotator for label $y$ of $\mathbf{x} \in \mathcal{A}_r$ using candidate set $\hat{Y}(\mathbf{x})$ to form $\mathcal{D}_r$.
6:     Get model $\boldsymbol{\theta}_r$ trained on $\bigcup_{i=0}^{r} \mathcal{D}_i$.
7: **end for**
8: **return** Final model $\boldsymbol{\theta}_R$.

---

In the following, we first introduce candidate set query (CSQ) and discuss its efficiency in labeling cost (Sec. 3.1). Then, we present a method to construct a candidate class set based on the prediction uncertainty of a trained model for a given sample (Sec. 3.2). Lastly, we introduce an acquisition function designed to consider cost efficiency as well as information gain (Sec. 3.3). The overall pipeline of the CSQ framework is summarized in Algorithm 1.

## 3.1 CANDIDATE SET QUERY

Candidate set query (CSQ) for an instance $\mathbf{x}$ is associated with a (non-empty) candidate set $\hat{Y}(\mathbf{x}) \subseteq \mathcal{Y}$ such that $1 \leq |\hat{Y}(\mathbf{x})| \leq L$. CSQ first asks the annotator to choose the ground-truth class in $\hat{Y}(\mathbf{x})$ (if exists) or to verify the absence of the ground-truth label in $\hat{Y}(\mathbf{x})$, *i.e.*, the annotator is first asked to pick an option out of $(k + 1)$ choices, where $k = |\hat{Y}(\mathbf{x})|$. Only if the absence of the ground-truth class in the candidate set is verified, the annotator is further asked to select the ground-truth class from the remaining ones $\mathcal{Y} \setminus \hat{Y}(\mathbf{x})$. To analyze the cost of CSQ, following the information-theoretic cost model (Hu et al., 2020) and our empirical study in Table. 1, we assume that the cost of choosing an option out of $k$ many candidates is $\log_2 k$. Then, the labeling cost $\Gamma(\mathbf{x}, y, \hat{Y}(\mathbf{x}))$ of CSQ for input $\mathbf{x}$, ground-truth label $y$, and candidate set $\hat{Y}(\mathbf{x})$ can be obtained as:

$$\Gamma(\mathbf{x}, y, \hat{Y}(\mathbf{x})) = \begin{cases} \log_2(k+1) & \text{if } y \in \hat{Y}(\mathbf{x}) \\ \log_2(k+1) + \log_2(L-k) & \text{otherwise} \end{cases}. \tag{1}$$

The conventional query in AL is a special case of CSQ where $\hat{Y}(\mathbf{x}) = \mathcal{Y}$, and it is inefficient since the annotator must search through the entire set of size $L$ with a cost of $\log_2 L$. The following theorem reveals the condition under which the expected cost of CSQ offers an improvement over that of the conventional query.

**Theorem 3.1.** *Assume the information-theoretic cost model (Hu et al., 2020) of selecting one out of $L$ possible options to be $\log_2 L$. Let $L \geq 2$ be the number of classes, $k = |\hat{Y}(\mathbf{x})|$, and $\alpha$ be the probability that the candidate set $\hat{Y}(\mathbf{x})$ does not include the ground-truth class of instance $\mathbf{x}$. For the expected cost of conventional query $C_{\text{con}}$ and that of candidate set query $C_{\text{csq}}$, if*

$$\frac{\log_2(k+1)}{\log_2 L} < 1 - \alpha, \tag{2}$$

*then $C_{\text{csq}}(L, \mathbf{x}, \alpha) < C_{\text{con}}(L, \mathbf{x})$.*

*Proof.* Recalling the definition of $\alpha$, we have $C_{\text{csq}}(L, \mathbf{x}, \alpha) = (1 - \alpha) \log_2(k + 1) + \alpha \{\log_2(k + 1) + \log_2(L - k)\}$ from Eq. (1). As $L - k < L$, the cost ratio of $C_{\text{csq}}(L, \mathbf{x}, \alpha)$ to $C_{\text{con}}(L, \mathbf{x})$ for instance $\mathbf{x}$ is induced as:

$$\frac{C_{\text{csq}}(L, \mathbf{x}, \alpha)}{C_{\text{con}}(L, \mathbf{x})} = \frac{\log_2(k+1) + \alpha \log_2(L-k)}{\log_2 L} < \frac{\log_2(k+1)}{\log_2 L} + \alpha. \tag{3}$$

$\square$

Although we adopt the cost model from Hu et al. (2020), Theorem 3.1 holds for any cost model that increases monotonically with the number of options.

**Remark 3.2.** *If we constrain all candidate set sizes $k$ to be fixed, then $1 - \alpha$ corresponds to the top-$k$ accuracy $p_k$ of the model. Therefore, when $p_k \geq \log_L(k+1)$, CSQ consistently offers an improvement over the conventional query. For example, in datasets such as CIFAR-10 ($L = 10$), CIFAR-100 ($L = 100$), and ImageNet ($L = 1000$), if the model has a top-1 accuracy (i.e., $k = 1$) of at least 30.1%, 15.1%, and 10.0% respectively, then CSQ always provides an improvement.*

The above proof and remark demonstrate that under moderate conditions, CSQ is more efficient than the conventional query. As described in Eq. (3), the cost of CSQ decreases as both $\alpha$ and $k$ become smaller. However, since $k$ and $\alpha$ are inversely related, balancing the trade-off between $\alpha$ and $k$ is essential to fully leverage CSQ. Also, fixing candidate set sizes as in Remark 3.2 is suboptimal because it does not consider the uncertainty of individual samples. In the following section, we introduce our candidate set construction method, which both reflects the uncertainty of each sample and automatically balances the trade-off between $\alpha$ and $k$.

### 3.2 CONSTRUCTION OF COST-EFFICIENT CANDIDATE SET

As shown in Eq. (1) and Theorem 3.1, a candidate set needs to be both small and accurate in covering the ground-truth class. To do so, we propose using conformal prediction (Romano et al., 2020) to get a reliable and cost-optimized prediction set using the trained model $\boldsymbol{\theta}$ of the previous round.

**Calibration set collection.** Conformal prediction requires a labeled set for calibration that has not been used during the model training phase; this set must follow the same distribution as the target data for prediction (Vovk et al., 1999; Angelopoulos et al., 2023). To achieve this, we randomly select $n_{\text{cal}}$ samples from the actively selected data $\mathcal{A}_r$ and annotate them within the given budget to form $\mathcal{D}_{\text{cal}} = \{(\mathbf{x}_i, y_i)\}_{i=1}^{n_{\text{cal}}}$. The calibration set $\mathcal{D}_{\text{cal}}$ is used for conformal prediction and candidate set optimization, which will be explained in the following sections. Note that $\mathcal{D}_{\text{cal}}$ also contributes to model training after candidate set construction.

**Candidate set construction from conformal prediction.** Using $\boldsymbol{\theta}$ from the previous round and calibration set $\mathcal{D}_{\text{cal}}$ randomly sampled from $\mathcal{A}_r$, we obtain the sequence of conformal scores $\mathbf{s} := \{s_i\}_{i \in [n_{\text{cal}}]}$, where $s_i := 1 - P_{\boldsymbol{\theta}}(y_i \mid \mathbf{x}_i)$ for $(\mathbf{x}_i, y_i) \in \mathcal{D}_{\text{cal}}$. Then we obtain the $(1 - \alpha)$ empirical quantile $\hat{Q}(\alpha)$ of $\mathbf{s}$, which is given as,

$$\hat{Q}(\alpha) := \min_{s \in \mathbf{s}} \left\{ s : \frac{1}{n_{\text{cal}}} \sum_{s' \in \mathbf{s}} \left( \mathbb{1}[s' \leq s] \right) \geq \frac{\lceil (n_{\text{cal}} + 1)(1 - \alpha) \rceil}{n_{\text{cal}}} \right\} , \tag{4}$$

where $\alpha \in (0, 1)$ is an error rate hyperparameter, $\lceil \cdot \rceil$ is a ceiling function and $\mathbb{1}[\cdot]$ is an indicator function. We note that $\hat{Q}(\alpha)$ indicates that at least $100 \times (1 - \alpha)\%$ of the scores $\mathbf{s}$ are smaller than $\hat{Q}(\alpha)$. Then, we define the candidate set for an unlabeled instance $\mathbf{x}$ as follows:

$$\hat{Y}_{\boldsymbol{\theta}}(\mathbf{x}, \alpha) = \left\{ y : P_{\boldsymbol{\theta}}(y|\mathbf{x}) \geq 1 - \hat{Q}(\alpha), \ y \in \mathcal{Y} \right\} . \tag{5}$$

Previous study (Vovk et al., 1999; Angelopoulos et al., 2023) proved that the presented candidate set includes the correct label with the probability greater than $1 - \alpha$, which is given as,

$$P\left( y \in \hat{Y}_{\boldsymbol{\theta}}(\mathbf{x}, \alpha) \right) \geq 1 - \alpha . \tag{6}$$

This candidate set design reflects the uncertainty of each sample and is tailored to the improved model across successive AL rounds. More detailed procedure of conformal prediction is explained in Sec. C.

**Cost-optimized candidate set construction.** Although conformal prediction aims at adjusting candidate set $\hat{Y}_{\boldsymbol{\theta}}(\mathbf{x}, \alpha)$ to fit the condition of $\alpha$ as in Eq. (6), it does not take into account the size $k$ of the candidate set. The efficiency of CSQ improves as both $\alpha$ and the candidate set size $k$ decrease, as shown in Eq. (3). Since $\alpha$ and $k$ are inversely related, finding an optimal hyperparameter $\alpha$ to reduce the labeling cost is not straightforward. Hence, we optimize $pha$ to minimize labeling cost for the calibration set $\mathcal{D}_{\text{cal}}$ for further improvement of CSQ efficiency. To be specific, $\alpha$ is optimized by

$$\alpha^* := \arg\min_{\alpha \in (0,1)} \sum_{(\mathbf{x}, y) \in \mathcal{D}_{\text{cal}}} \Gamma(\mathbf{x}, y, \hat{Y}_{\boldsymbol{\theta}}(\mathbf{x}, \alpha)) , \tag{7}$$

where $\Gamma(\mathbf{x}, y, \hat{Y}_{\boldsymbol{\theta}}(\mathbf{x}, \alpha))$ is the labeling cost in Eq. (1). By optimizing $\alpha$ in this way, we utilize conformal prediction to construct candidate sets in a more cost-efficient manner, as the error rate is tailored to minimize the expected labeling cost for each round. Notably, if we define the corner case $\hat{Y}_{\boldsymbol{\theta}}(\mathbf{x}, 0) = \mathcal{Y}$, CSQ includes the conventional query at $\alpha = 0$ within the search space for $\alpha^*$. This makes CSQ is at least as efficient as, and often more efficient than, the conventional query.

Note that to construct the candidate set query, the calibration set $\mathcal{D}_{\text{cal}}$ is required to calculate $(1-\alpha^*)$ quantile in Eq. (4). Thus, when getting annotations of $\mathcal{D}_{\text{cal}}$ in the calibration set collection step, candidate set query of the current round cannot be applied. To avoid this circular dependency, the quantile from the previous round is used when labeling $\mathcal{D}_{\text{cal}}$.

## 3.3 Cost-efficient acquisition function

Since the labeling cost of each sample varies in CSQ, we propose to consider the cost for active sampling. We implement an acquisition function that evaluates samples based on the ratio of the estimated information gain to the estimated labeling cost. The information gain is quantified using one of the well-established acquisition scores from prior research. Specifically, we adopt methods such as BADGE (Ash et al., 2020) and entropy, although our approach can incorporate any acquisition scoring function. Given a conventional acquisition score $g_{\text{score}}(\mathbf{x})$, the proposed cost-efficient acquisition function $g_{\text{cost}}$ is given as,

$$g_{\text{cost}}(\mathbf{x}) := \frac{(1 + g_{\text{score}}(\mathbf{x}))^d}{\log_2(k + 1) + \alpha^* \log_2(L - k)} , \tag{8}$$

where $d$ is a hyperparameter adjusting the influence of $g_{\text{score}}(\mathbf{x})$ and $\alpha^*$ is an optimized error rate hyperparameter obtained by Eq. (7). The denominator is an expected cost derived from our cost model (Eq. (1)), considering two cases: the correct label is included or excluded from the candidate set, which is $(1 - \alpha^*) \log_2(k + 1) + \alpha^* \{\log_2(k + 1) + \log_2(L - k)\}$. This expected cost assumes the candidate set to include the ground-truth class with probability of $1 - \alpha^*$, which is supported by the coverage guarantee in Eq. (6). We normalize $g_{\text{score}}$ to $[0, 1]$, as any existing acquisition score can be employed for $g_{\text{score}}(x)$.

## 4 Experiments

### 4.1 Experimental setup

**Datasets.** Our method is evaluated using three image classification datasets: CIFAR-10 (Krizhevsky et al., 2009), CIFAR-100 (Krizhevsky et al., 2009), and ImageNet64x64 (Chrabaszcz et al., 2017). CIFAR-10 comprises 50K training and 10K validation images across 10 classes. CIFAR-100 contains the same number of images as CIFAR-10, but with 100 classes. ImageNet64x64 is a downsampled version of ImageNet (Deng et al., 2009a) with a resolution of $64 \times 64$, which consists of 1.2M training and 50K validation images with 1000 classes. Following previous studies, we evaluate a model using the validation split of each dataset.

**Implementation details.** For CIFAR-10 and CIFAR-100, we adopt ResNet-18 (He et al., 2016) as a classification model. We train it for 200 epochs using AdamW (Loshchilov & Hutter, 2019) optimizer with an initial learning rate of $1\text{e}{-}3$, decreasing by a factor of 0.2 at epochs 60, 120, and 160. We apply a weight decay of $5\text{e}{-}4$ and a data augmentation consists of random crop, random horizontal flip, and random rotation. For ImageNet64x64, we adopt WRN-36-5 (Zagoruyko, 2016), and train it for 30 epochs using AdamW optimizer with an initial learning rate of $8\text{e}{-}3$. We apply a learning rate warm-up for 10 epochs from $2\text{e}{-}3$. After the warm-up, we decay the learning rate by a factor of 0.2 every 10 epochs. We adopt random horizontal flip and random translation as data augmentation. For all the datasets, we use Mix-up (Zhang et al., 2018), where a mixing ratio is sampled from Beta$(1, 1)$. The hyperparameter $d$ in Eq. (8) is set to 1.0, 0.5, and 1.2 for cost-efficient entropy sampling on CIFAR-10, CIFAR-100, and ImageNet64x64, respectively. For cost-efficient BADGE sampling, $d$ is set to 1.1 for CIFAR-10 and 1.2 for CIFAR-100. Also, we set the size of calibration dataset $n_{\text{cal}}$ to 500 for CIFAR-10 and CIFAR-100, and 5K for ImageNet64x64.

**Active learning protocol.** For CIFAR-10, we conduct 10 AL rounds of consecutive data sampling and model updates, while for CIFAR-100, we perform 9 AL rounds. In both cases, the per-round

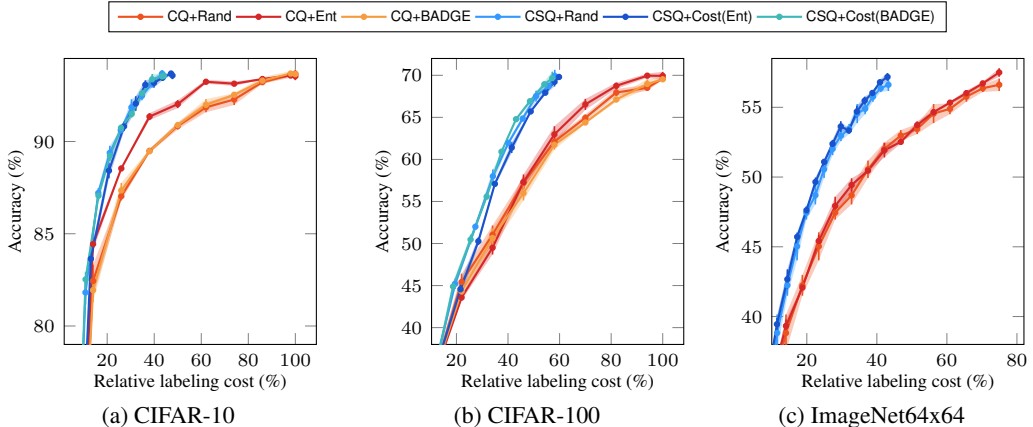

Figure 2: Accuracy (%) versus relative labeling cost (%) for conventional query (CQ) and candidate set query (CSQ) with different acquisition functions: CQ using Random, Entropy, and BADGE, and CSQ using Random and cost-efficient sampling. CSQ approches (blue lines) consistently outperforms the CQ baselines (red lines) by a significant margin across various budgets, acquisition functions, and datasets.

Table 1: The results of the user study showing the annotation time (second) and accuracy (%) for the same images with varying size of class options (candidate set). This result demonstrates that a small candidate set improves both labeling efficiency and accuracy.

| Size of candidate set | 4 | 8 | 16 | 32 |
|---|---|---|---|---|
| Annotation time (s) | $\mathbf{69.4}_{\pm 13.8}$ | $91.5_{\pm 27.3}$ | $116.9_{\pm 29.6}$ | $166.9_{\pm 30.8}$ |
| Accuracy (%) | $\mathbf{100.0}_{\pm 0.0}$ | $98.5_{\pm 3.2}$ | $99.5_{\pm 1.5}$ | $95.5_{\pm 5.2}$ |

budget is 6K images. For ImageNet64x64, we conduct 16 AL rounds with a per-round budget of 60K images. The detailed budget configuration for the three datasets is shown in Table 3. In the initial round, we randomly sample 1K images for CIFAR-10, 5K images for CIFAR-100, and 60K images for ImageNet64x64. In each round, the model is evaluated based on two factors: its accuracy (%) on the validation set, and the annotation cost required to train it. The annotation cost is defined as a relative labeling cost (%) compared to the cost of labeling the entire training set using the conventional query, given by $N \log_2 L$, where $N$ is the size of the entire training set, and $L$ is the number of classes. We conduct all experiments with three independent trials with different random seeds and report the mean and standard deviation to ensure reproducibility.

**Baseline methods.** We compare the proposed candidate set query (CSQ) with the conventional query (CQ) in combination with various sampling strategies. Following the established sampling strategies in previous AL studies, we employ random sampling (Rand), entropy-based sampling (Ent), and BADGE sampling (BADGE) (Ash et al., 2020). Cost(Ent) indicates the proposed cost-efficient sampling (Eq. (8)) combined with the entropy acquisition function, and Cost(BADGE) is the one combined with BADGE. We denote the combination of the query and sampling method with '+', *i.e.*, CSQ+Rand is a candidate set query with random sampling.

## 4.2 EXPERIMENTAL RESULTS

**Candidate set query vs. Conventional query.** In Fig. 2, we compare the performance of candidate set query (CSQ) with the conventional query (CQ) on CIFAR-10, CIFAR-100, and ImageNet64x64 with different acquisition functions. CSQ approaches consistently outperforms the CQ approaches across various acquisition functions and datasets, demonstrating the general effectiveness of our method. Notably, CSQ reduce the labeling cost of CQ by 56%, 43%, and 42% CIFAR-10, CIFAR-100, and ImageNet64x64, respectively. This is promising as it shows that the same volume of labeled data can be obtained at roughly half the cost, without introducing any label noise or sample bias.

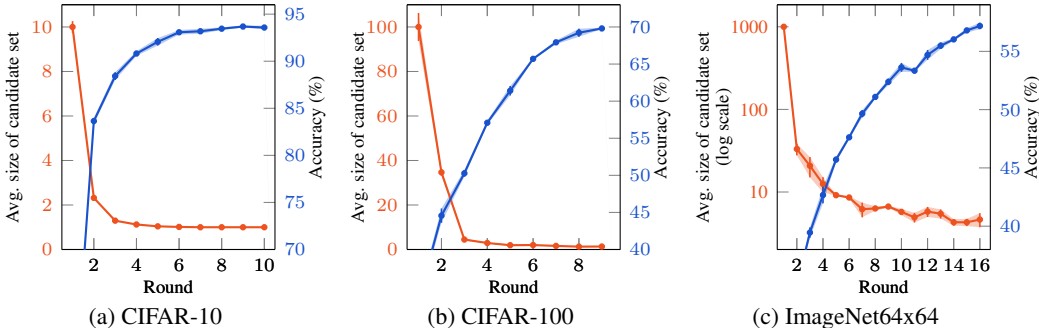

Figure 3: Average size of the candidate set and accuracy (%) of our method with cost-efficient entropy sampling in varying rounds on CIFAR-10, CIFAR-100, and ImageNet64x64. Our candidate set design adapts to the increasing accuracy of the model over successive AL rounds, reducing it as the model improves.

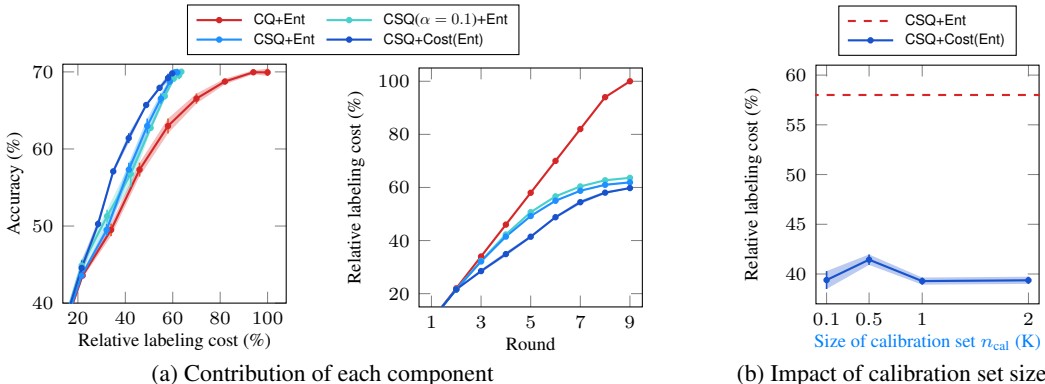

Figure 4: (a) Contribution of each component of our method, measured by Accuracy (%) versus relative labeling cost (%) (*left*), and relative labeling cost (%) versus AL rounds (*right*) on CIFAR-100. The result compare the full method (CSQ+Cost(Ent)), the method without acquisition function in Eq. (8) (CSQ+Ent), without $\alpha$ optimization in Eq. (7), where $\alpha$ is fixed to 0.1 (CSQ(Fixed $\alpha$)+Ent), and without CSQ (CQ+Ent). All components of our method lead to steady performance improvement over varying rounds. (b) Relative labeling cost (%) at fifth round with varying calibration set sizes $n_{cal}$ in Eq. (4) on CIFAR-100. The dashed line indicates the relative labeling cost (%) of the baseline (CQ+Ent). Our method demonstrates robustness to the change in calibration set size.

Notably, the performance gain of CSQ increases as the model improves, as it is tailored to improved model.

**Empirical evidence for Theorem 3.1.** We empirically demonstrate that the conditions for Theorem 3.1 are met. First, we verify the information-theoretic annotation cost assumption through a user study with 40 annotators. Each group of 10 annotators labels 20 queries with candidate set sizes of 4, 8, 16, and 32. Details are provided in Appendix A. Table 1 shows that smaller candidate sets improve both labeling efficiency and accuracy. The results also align closely with theoretical costs, as shown in Fig. 1(*right*). Next, we demonstrate that the proposed CSQ effectively reduces both the candidate set size $k$ and error rate $\alpha$ throughout the AL rounds. As shown in Fig. 3b, after the first round, CSQ achieves a sufficiently small $k$ and continues to reduce it as accuracy improves.

### 4.3 Ablation Studies

**Contribution of each component.** Figure 4a demonstrates the contribution of each component in our method across varying AL rounds: candidate set query (Eq. (5)), cost optimization of $\alpha$ (Eq. (7)), and the proposed acquisition function (Eq. (8)). The results show consistent performance improvements from each component in every round. The performance gap between CQ+Ent and CSQ($\alpha = 0.1$)+Ent verifies the efficacy of proposed CSQ framework, which provides the largest

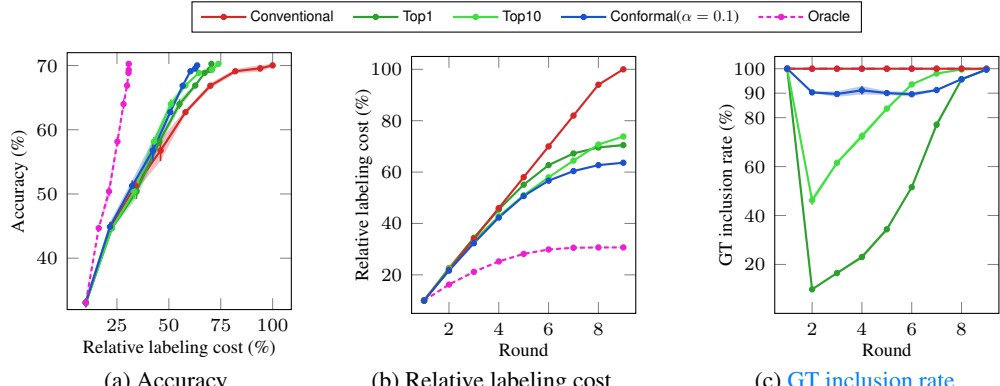

(a) Accuracy      (b) Relative labeling cost      (c) GT inclusion rate

Figure 5: Impact of the candidate set design evaluated on CIFAR-100 using conventional query with all classes (Conventional), top-1 prediction from model (Top1), top-10 prediction from model (Top10), our method with conformal prediction with fixed $\alpha = 0.1$ (Conformal($\alpha = 0.1$)), and the smallest top-k prediction sets always including ground-truth class (Oracle). For comparison, the same entropy sampling is used, ensuring that while labeling costs vary, the accuracy per round remains consistent to isolate the effect of candidate set design. (a) The proposed method constantly outperforms the baselines in accuracy (%) relative to labeling cost (%). (b) Our design achieves greater reduction in labeling cost compared to baselines. (c) Our candidate set effectively includes the ground-truth class in over 90% of cases ($= 1 - \alpha$), even when model accuracy low.

improvement. The gap between CSQ($\alpha = 0.1$)+Ent and CSQ+Ent shows the impact of $\alpha$ optimization, offering modest but steady gains across rounds. Finally, the gap between CSQ+Ent and CSQ+Cost(Ent) shows the effectiveness of our acquisition function, particularly from 4 to 6 rounds.

**Impact of calibration set size.** In Fig. 4b, we evaluate the relative labeling cost (%) at the fifth round with varying calibration set sizes $n_{cal}$ in Eq. (4) to assess its impact on the performance on CIFAR-100. A larger $n_{cal}$ may improve the accuracy of conformal prediction and $\alpha$ optimization but is less efficient in terms of labeling cost. As shown in Fig. 4b our method shows robust performance, only varying less than 2%p as the calibration set size changes from 0.1K to 2K. Even with a calibration set size of just 100, our method significantly outperforms the baseline reducing the cost by 18%p.

**Impact of conformal prediction for candidate set design.** Figure 5 illustrates the effectiveness of conformal prediction (Conformal ($\alpha = 0.1$)) for candidate set construction on CIFAR-100, compared to baselines: Conventional (using all classes), Top1 (top-1 prediction), Top10 (top-10 predictions), and Oracle (smallest top-$k$ set always containing the ground truth). Note that Oracle represents an unattainable upper bound requiring knowledge of the ground truth. For consistency, we fixed $\alpha = 0.1$ in Eq. (5). Figures 5a and 5b show that conformal prediction consistently reduces labeling cost compared to the baselines. While Top10 is effective in the early rounds and Top1 becomes more efficient as the model improves, our method adapts throughout and outperforms all baselines in every round. Figure 5c demonstrates that with $\alpha = 0.1$, our method includes the ground-truth class in over 90% of cases, aligning with Eq. (6), while the top-$k$ baselines show lower inclusion rates, especially in early and middle rounds. This demonstrates that conformal prediction effectively adjusts candidate set sizes based on sample uncertainty, ensuring ground-truth inclusion and improving labeling efficiency.

**Impact of cost-optimized candidate set construction.** In Fig. 6, we present the impact of cost-optimized candidate set construction as in Eq. (7), evaluated on CIFAR-100 using entropy sampling, in terms of relative labeling cost (%). As shown in Fig. 6a, the proposed optimization consistently reduces labeling cost across all rounds by selecting the optimal $\alpha = \alpha^*$. In Fig. 6b, the magenta diamonds indicate how the most cost-effective $\alpha$ changes with each active learning round, showing that labeling costs vary significantly depending on the chosen $\alpha$. Our method enhances cost efficiency by selecting the optimal $\alpha^*$ (cyan diamonds) in each round through cost optimization, leading to more efficient candidate sets.

**Qualitative result of constructed candidate sets.** In Fig. 7, we present qualitative results showing input images and their corresponding candidate sets on ImageNet64x64. Thanks to the conformal

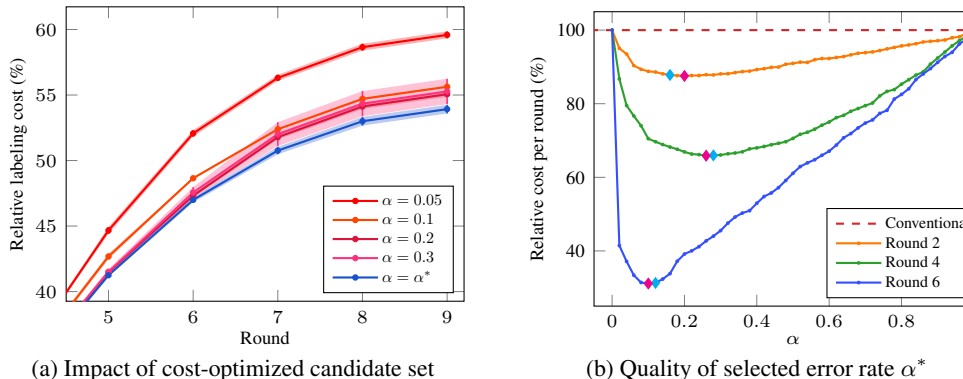

(a) Impact of cost-optimized candidate set

(b) Quality of selected error rate $\alpha^*$

Figure 6: Impact of cost-optimized candidate set construction as in Eq. (7), evaluated on CIFAR-100 with entropy sampling. (a) Relative labeling cost (%) versus AL rounds with different error rate $\alpha$ and the $\alpha^*$ selected by the proposed cost optimization (Eq. (7)). (b) Relative labeling cost per round (%) versus $\alpha$ across varying AL rounds. Labeling cost is measured as the ratio compared to labeling all images in a single round using the conventional query. The magenta diamond represents the true optimal $\alpha$ minimizing the cost for sampled data, while the cyan diamond represents the $\alpha^*$ selected from Eq. (7). The dashed line indicates the baseline cost from the conventional query.

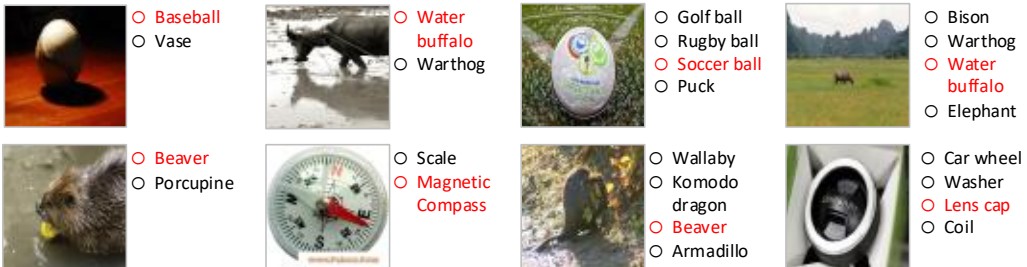

Figure 7: Qualitative results of input images and their corresponding candidate sets constructed from our method in fifth round on ImageNet64x64. The ground-truth class is highlighted in red (best viewed in color).

prediction, the proposed method allows for flexible adjustment of the candidate set for each sample. For certain samples (Fig. 7(*left*)), the candidate set is reduced to minimize labeling cost, while for uncertain samples (Fig. 7(*right*)), the candidate set is expanded to include the true label.

## 5 CONCLUSION

We propose candidate set query (CSQ), a cost-efficient active learning framework for classification. By narrowing down candidates likely to include the ground-truth class, our approach significantly reduces labeling costs. To manage varying candidate set sizes, we introduce a novel acquisition function that balances performance gain with labeling cost. Experiments on CIFAR-10, CIFAR-100, and ImageNet64x64 show that CSQ significantly reduces labeling costs, demonstrating its potential for efficiently scaling large annotated datasets.

**Limitation and Future work.** One limitation is that the proposed acquisition function lacks theoretical guarantee for label complexity (Dasgupta, 2011; Hanneke et al., 2014) at this point. Establishing a theoretical understanding to quantify the cost required to achieve a target performance remains an interesting direction for future work. Also, although our acquisition function shows improvements over baselines, it relies on hyperparameter $d$ to balance the trade-off between cost and informativeness. If $g_{\text{score}}(\mathbf{x})$ could measure the true influence (Koh & Liang, 2017) on accuracy, setting $d = 1$ in Eq. (8) would optimize cost per influence, potentially yielding an optimal acquisition function. However, improving $g_{\text{score}}(\mathbf{x})$ is beyond the scope of this work.

## 6 REPRODUCIBILITY STATEMENT

We have included the source code for our experiments as part of the supplementary material. Detailed instructions on loading datasets and running the code to reproduce the experiment results are provided in Appendix B. The training configurations, active learning settings, and hyperparameter details are discussed in Sec. 4.1.

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

# A   DETAILS OF USER STUDY

Q. Select the class that corresponds to the image.

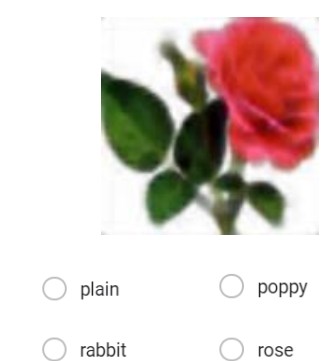

○ plain          ○ poppy

○ rabbit         ○ rose

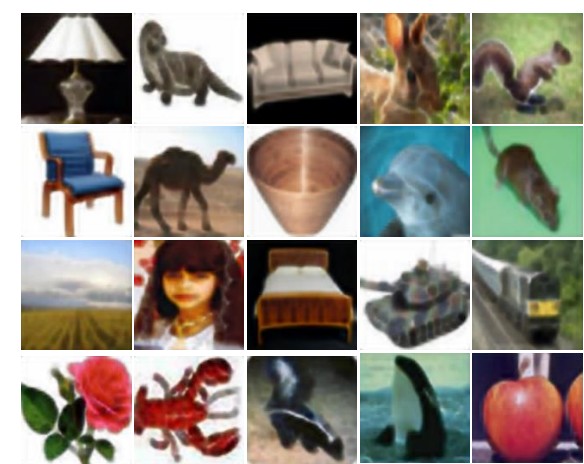

(a) Questionnaire with four candidates ($k = 4$)          (b) Example queries in CIFAR-100

Figure 8: Questionnaire and examples used in the user study. (a) Each question contains an instruction, an image, and a set of candidates. In this case, the candidate set size is 4. (b) We utilize 20 images in CIFAR-100, each with a resolution of 128 x 128 pixels.

We conduct a user study to examine how the size of a candidate set, $k$ in Sec. 3.1, affects the annotation time in practice. Figure 8 presents examples of the questionnaires and all images used in our user study. To facilitate easy comparison with the theoretical costs (Hu et al., 2018), we set the candidate set sizes to 4, 8, 16, and 32. To be specific about Figure 8, we use CIFAR-100 images resized to $128 \times 128$ using super resolution[1] to enhance visibility for annotators. We first randomly select 20 classes in CIFAR-100 and choose one image per class to organize the questionnaires. For small-sized candidate sets, we ensure the inclusion of the ground truth by randomly trimming around it when generating the candidate sets.

We divide 44 annotators into four groups of 11 for each candidate set size to perform labeling tasks. To account for potential outliers, we exclude the results of the annotators whose time taken deviates the most from the average time in each group. Table 2 shows that as the candidate set size increases, the time per query increases and the accuracy decreases. In addition, on the right side of Table 2, a comparison between the experimental costs and theoretical costs reveals a significant correlation of 0.97.

Table 2: User study for different sizes of candidate set query.

| $k$ | Total time (s) | Time per query (s) | Accuracy (%) | Experimental | Theoretical |
|---|---|---|---|---|---|
| 4 | $\mathbf{69.4}_{\pm 13.8}$ | $\mathbf{3.47}_{\pm 0.69}$ | $\mathbf{100.0}_{\pm 0.0}$ | 2.0 | 2 |
| 8 | $91.5_{\pm 27.3}$ | $5.20_{\pm 1.36}$ | $98.5_{\pm 3.2}$ | 2.6 | 3 |
| 16 | $116.9_{\pm 29.6}$ | $6.94_{\pm 1.48}$ | $99.5_{\pm 1.5}$ | 3.4 | 4 |
| 32 | $166.9_{\pm 30.8}$ | $8.35_{\pm 1.54}$ | $95.5_{\pm 5.2}$ | 4.8 | 5 |

# B   IMPLEMENTATION DETAILS AND CONFIGURATION

Table 3 presents the configuration of our main experiments for each dataset. In all experiments, we fixed the per-round budget, which limits the number of annotated instances per active learning (AL) round. Given this budget constraint, we compute the labeling cost for each AL round to assess labeling efficiency." The batch size for CIFAR-10 and CIFAR-100 was determined to 128, while that for ImageNet64x64 is set to 128. We normalized the input image to ensure the stability of the

---

[1]https://www.kaggle.com/datasets/joaopauloschuler/cifar100-128x128-resized-via-cai-super-resolution

training. We trained our classification model on CIFAR-10 and CIFAR-100 using NVIDIA RTX 3090 and on ImageNet64x64 using 4 NVIDIA A100 GPUs in parallel. The training requires about 5 GPU hours for CIFAR-10 and CIFAR-100, and about 1.5 GPU days for ImageNet64x64.

Table 3: Detailed dataset and budget configuration for the proposed scenario.

| Dataset | $L$ | $\log_2 L$ | Size | Cost of full label | # of rounds | Per-round budget |
|---|---|---|---|---|---|---|
| CIFAR-10 | 10 | 3.322 | 50K | 166.1K | 10 | 6K |
| CIFAR-100 | 100 | 6.644 | 50K | 332.2K | 9 | 6K |
| ImageNet64x64 | 1000 | 9.966 | 1.2M | 12.7M | 16 | 60K |

**Code.** This part demonstrates the reproducibility of our work by providing comprehensive details on the source code release. We have made available the entire framework, which includes the data sampling method, evaluation procedures, and the overall training pipeline. Our aim is to ensure that other researchers can easily replicate and build upon our results. To get started with running the code, please refer to the `script.sh` file. This script contains the necessary commands and instructions to execute our experiments seamlessly. To better understand our proposed method, you can examine the Python script `al/strategy_dtopk.py`. This file includes the implementation details of our active learning strategies, particularly *candidate set suery* design. Furthermore, our code can run on CIFAR-10, CIFAR-100 [2], and ImageNet64x64 [3], which are available online. Note that you can modify the running configuration such as dataset, sampling method, and budget through command-line arguments.

## C ADDITIONAL CLARIFICATION ON CANDIDATE SET CONSTRUCTION

**The detailed procedure of computing $\hat{Q}(\alpha)$ in Eq. (4).** We begin with computing the conformal scores $\mathbf{s}$ for the calibration dataset $\mathcal{D}_{\text{cal}}$. For each data point $(\mathbf{x}_i, y_i) \in \mathcal{D}_{\text{cal}}$, the conformal score is defined as:

$$s_i := 1 - P_{\boldsymbol{\theta}}(y_i \mid \mathbf{x}_i), \quad \text{for } i = 1, 2, \cdots, n_{\text{cal}}, \tag{9}$$

where $n_{\text{cal}} = |\mathcal{D}_{\text{cal}}|$. Using these scores, we define the empirical distribution function $F_n(s)$, which measures the proportion of scores less than or equal to a given value $s$. Formally, $F_n(s)$ is expressed as:

$$F_n(s) = \frac{1}{n_{\text{cal}}} \sum_{i=1}^{n_{\text{cal}}} \mathbb{1}[s_i \leq s], \tag{10}$$

where $\mathbb{1}[\cdot]$ is an indicator function. The $(1 - \alpha)$ empirical quantile is then defined as the smallest score $s_i$ such that the proportion of scores satisfying $s_i \leq s$ is at least $(1 - \alpha)$. Mathematically, this is given as $\min_{i \in [n_{\text{cal}}]} \{F_n(s_i) \geq 1 - \alpha\}$, where $[n_{\text{cal}}] = \{1, 2, \cdots, n_{\text{cal}}\}$. To ensure robustness under limited sample sizes, we adjust $(1 - \alpha)$ into $\lceil (n_{\text{cal}} + 1)(1 - \alpha) \rceil / n_{\text{cal}}$ when defining $\hat{Q}(\alpha)$, which is defined as:

$$\hat{Q}(\alpha) := \min_{i \in [n_{\text{cal}}]} \left\{ F_n(s_i) \geq \frac{\lceil (n_{\text{cal}} + 1)(1 - \alpha) \rceil}{n_{\text{cal}}} \right\}. \tag{11}$$

Note that Eq. (11) is equivalent to Eq. (4).

## D DISCUSSION ON HANDLING OUTLIERS AND ANOMALOUS DATAPOINTS

Dealing with out-of-distribution (OOD) data points showing high uncertainty scores has been a chronic issue in active learning and may affect the efficiency of candidate set query (CSQ). Recent open-set active learning approaches (Du et al., 2021; Kothawade et al., 2021; Ning et al., 2022; Park et al., 2022; Yang et al., 2024) tackle this by filtering out OOD samples during active sampling

---

[2] https://www.cs.toronto.edu/~kriz/cifar.html
[3] https://patrykchrabaszcz.github.io/Imagenet32/

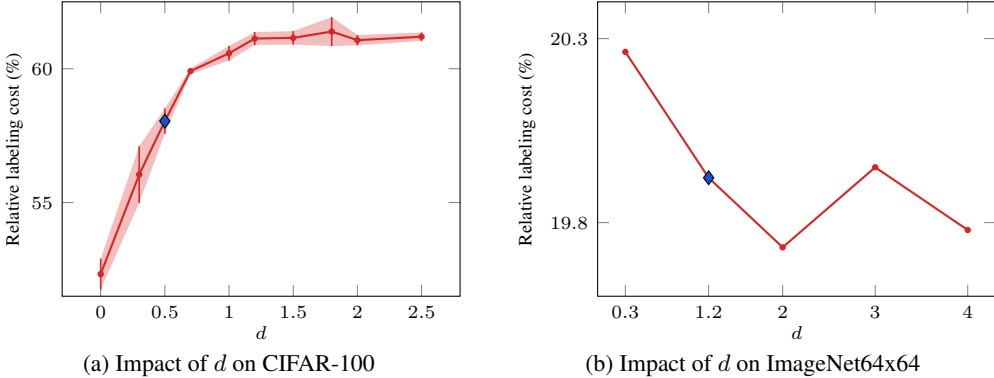

Figure 9: Sensitivity of the labeling cost to the hyperparameter $d$ in Eq. (8), evaluated on CIFAR-100 and ImageNet64x64 with CSQ+Cost(Ent). We report the relative labeling cost (%) for various values of $d$ at a specific active learning round. The blue diamond marks the $d$ value used in the main experiments. (a) Results for CIFAR-100 at the eighth round. (b) Results for ImageNet64x64 at the sixth round. We report results for ImageNet64x64 using only a single random seed.

using an OOD classifier. Our CSQ framework integrates seamlessly with these methods, focusing on labeling in-distribution (ID) samples to prevent cost inefficiencies.

However, as OOD classifiers are not flawless, some OOD samples may still be selected. One advantage of our method is its ability to leverage the calibration set to capture information about such mixed OOD samples. This enables adjustments such as increasing the OOD classifier threshold to exclude more OOD-like data or incorporating the OOD ratio into the alpha optimization process in Eq. (7). Optimizing the combination of OOD and ID classifier scores within the calibration set or designing better OOD-aware queries presents promising future research directions.

## E    IMPACT OF HYPERPARAMETER $d$

**Impact of informativeness-cost balancing hyperparameter $d$.** The hyperparameter $d$ in our acquisition function (Eq. (8)) balances the trade-off between labeling cost and the informativeness of a selected sample, requiring both factors to be considered. We provide a comprehensive analysis showing the trend of performance in accuracy with varying $d$ values over AL rounds for CIFAR-10, CIFAR-100, and ImageNet64x64 in Fig. 10. In CIFAR-10 (Fig. 10a), both accuracy and labeling cost remain robust to the change of $d$, varying only 0.5%p in accuracy. In CIFAR-100 (Fig. 10b), the overall performance is still insensitive yet slightly increasing as $d$ decreases. On the other hand, in ImageNet64x64 (Fig. 10c), the performance decreases as $d$ increases until it reaches 2.0. Regarding that a larger $d$ prioritizes more uncertain samples, this result aligns with recent observations that uncertainty-based selection performs better in scenarios with larger labeling budgets (Hacohen et al., 2022).

**Guidelines for selecting proper hyperparameter $d$.** We provide the following guidelines for setting $d$. For datasets with fewer than 100 classes, $d$ values between 0.3 and 1.0 may be effective, as they ensure robustness on simple datasets like CIFAR-10 and reduce labeling costs on more complex datasets like CIFAR-100. For larger datasets closer in scale to ImageNet, exploring $d \geq 1.0$ can help further improve the model performance.

## F    COMPARISON WITH SIFTING OUT BASELINE FOR CANDIDATE SET CONSTRUCTION

Figure 11 compares the candidate set construction method of our candidate set query (CSQ) with a baseline (CSQ-sift) that sifts out classes with softmax values below $0.1 \times 1/C$, where $C$ is the number of classes, across AL rounds, using entropy and BADGE (Ash et al., 2020) sampling on CIFAR-100. The results show that CSQ is more cost-efficient, reducing relative labeling cost by 7.2%p compared to CSQ-sift at the ninth round even with entropy sampling, favoring samples with

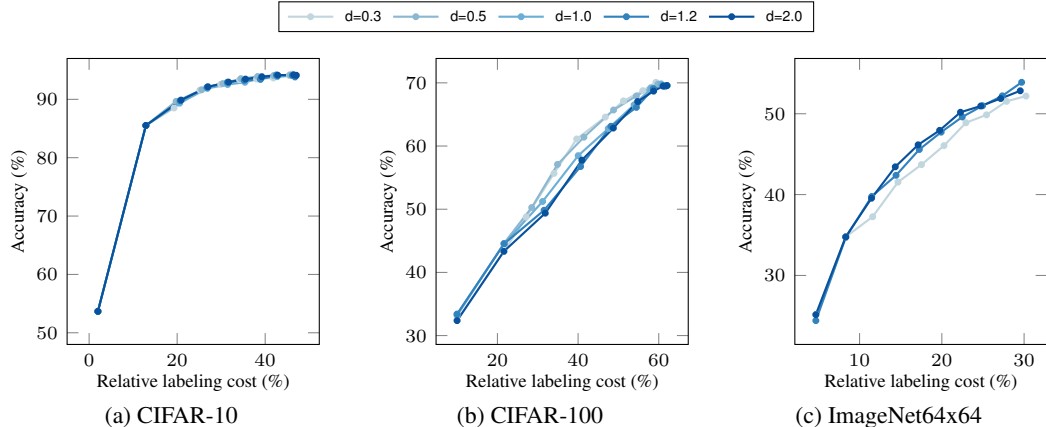

(a) CIFAR-10     (b) CIFAR-100     (c) ImageNet64x64

Figure 10: Accuracy (%) versus relative labeling cost (%) with varying hyperparameter $d$ in Eq. (8) across AL rounds, evaluated on CIFAR-10, CIFAR-100 and ImageNet64x64 with CSQ+Cost(Ent). For our main experiments, we set $d = 1.0$, $d = 0.5$, and $d = 1.2$, for CIFAR-10, CIFAR-100, and ImageNet64x64, respectively.

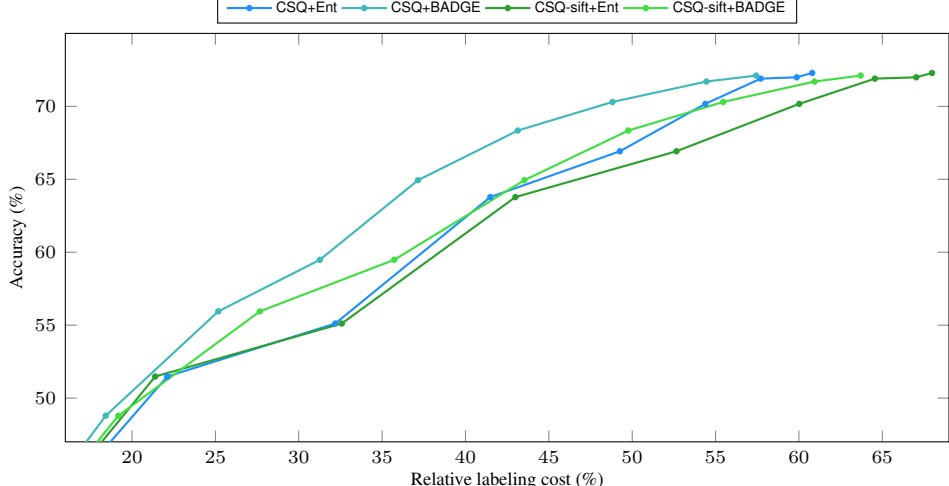

Figure 11: Accuracy (%) versus relative labeling cost (%) for candidate set query (CSQ) and baseline that sifts out classes with softmax values below $0.1 \times 1/C$ ($C$: number of classes, CSQ-sift), using Entropy and BADGE sampling. CSQ approches (blue lines) consistently outperforms the CSQ-sift baselines (green lines) across various budgets and acquisition functions.

uniform softmax values. When paired with BADGE, a more advanced diversity-aware acquisition function, CSQ shows additional cost savings.

CSQ also offers a key advantage over the heuristic variant (CSQ-sift) by providing a theoretical guarantee of including the correct class, enabling the use of our acquisition function. This acquisition function further enhances cost-efficiency.

## G   COMPATIBILITY BETWEEN CANDIDATE SET CONSTRUCTION AND UNCERTAIN SAMPLES

Figure 12 compares CSQ and conventional query (CQ) on CIFAR-100 with entropy-based sampling (Ent) and our acquisition function with entropy measure (Cost(Ent), Eq. (8)) across AL rounds, with a fixed number of samples per round.

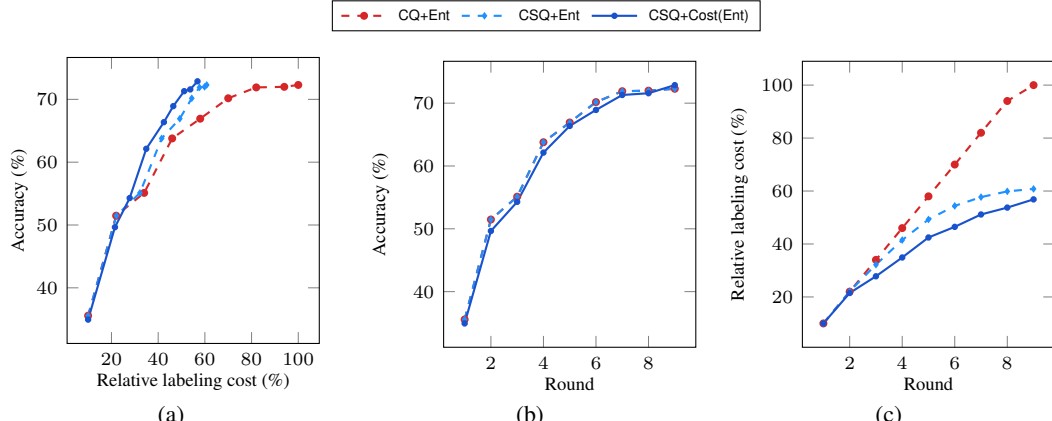

Figure 12: Comparison of candidate set query (CSQ) and conventional query (CQ) on CIFAR-100 with entropy sampling (Ent) and cost-efficient entropy sampling (Cost(Ent)) varying AL rounds. A fixed number of samples are selected at each AL round. (a) Accuracy (%) versus relative labeling cost (%) showing the accuracy per cost. (b) Accuracy (%) versus AL rounds showing the accuracy varies with the number of samples. Note that the lines of CQ+Ent and CSQ+Ent completely overlap, as they use the same sampling method. (c) Relative labeling cost (%) versus AL rounds.

**Our acquisition function provides superior accuracy per cost.** The comparison between CSQ+Cost(Ent) and CSQ+Ent demonstrates that the proposed acquisition function reduces labeling costs with only a marginal accuracy trade-off.

**Candidate set query (CSQ) can reduce labeling costs even for uncertain samples.** The comparison between CQ+Ent and CSQ+Ent demonstrates that CSQ effectively reduces labeling costs, even with uncertainty-based sampling methods like entropy sampling. This shows that CSQ can narrow down annotation options even for uncertain samples. Note that CSQ+Ent shows the same accuracy as CQ+Ent, since they used the same sampling method.

## H   EXPERIMENTS IN LANGUAGE DOMAIN

**Dataset.** The R52 dataset (Lewis, 1997) is a subset of the Reuters-21578 (Lewis, 1997) news collection, specifically curated for text classification tasks. It comprises documents categorized into 52 distinct classes, with a total of 9,130 documents. The dataset is divided into 6,560 training documents and 2,570 testing documents. Each document is labeled with a single category, and the categories are selected to ensure that each has at least one document in both the training and testing sets. This structure makes the R52 dataset particularly suitable for evaluating text classification models.

**Implementation details.** We adopt an SVM model (Cortes, 1995) with sigmoid kernel for classification. We conduct 11 AL rounds of consecutive data sampling and model updates, where the per-round budget is 600. The hyperparameter $d$ for our acquisition function is set as 1.2. In the initial round, we randomly sample 300 samples. In each round, the model is evaluated based on three factors: its accuracy (%) and Micro-F1 (%).

Figure 13 presents a comparison of candidate set query (CSQ) and conventional query (CQ) on the text classification dataset (R52) with random sampling (Rand), entropy sampling (Ent), and our acquisition function with entropy measure (Cost(Ent), Eq. (8)) across AL rounds. CSQ approaches consistently outperform the CQ baselines by a significant margin across various budgets and acquisition functions. Especially at round 10, CSQ+Rand reduces labeling cost by 65.6%p compared to its conventional query baseline. The result demonstrates that the proposed CSQ framework generalizes to the text classification domain.

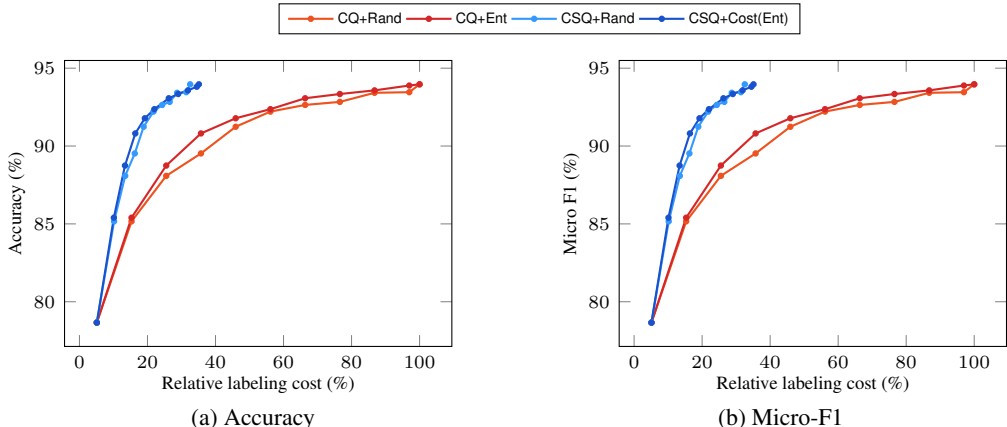

(a) Accuracy              (b) Micro-F1

Figure 13: Comparison between conventional query (CQ) and candidate set query (CSQ) with random sampling (Rand), entropy sampling (Ent), and cost-efficient entropy sampling (Cost(Ent) on text classification task with R52 dataset. (a) Accuracy (%) versus relative labeling cost (%). (b) Micro-F1 (%) versus relative labeling cost (%). CSQ approches (blue lines) consistently outperform the CQ baselines (red lines) by a significant margin across various budgets and acquisition functions.

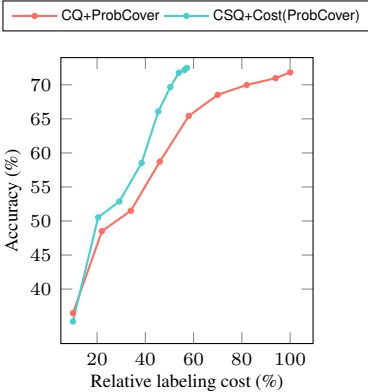

Figure 14: Comparison between conventional query (CQ) and candidate set query (CSQ) with Prob-Cover sampling (ProbCover) and cost-efficient ProbCover sampling (Cost(ProbCover) on CIFAR-100 dataset with AL rounds. CSQ approches (blue lines) consistently outperform the CQ baselines (red lines) by a significant margin across various budgets.

## I CANDIDATE SET QUERY PAIRED WITH ADVANCED AL ACQUISITION FUNCTIONS

We present additional experiments using ProbCover (Yehuda et al., 2022) sampling. ProbCover leverages self-supervised features for the entire training dataset to construct a weighted digraph, where the edge weights represent pairwise distances. It selects the sample with the highest out-degree for annotation. When the graph is depleted, it switches to random sampling from the unlabeled pool.

Figure 14 compares CSQ and CQ on CIFAR-100 with ProbCover sampling and cost-efficient Prob-Cover sampling (Cost(ProbCover)), across AL rounds. CSQ approaches consistently outperform the CQ baselines across various budgets and acquisition functions. In particular, the proposed method reduces labeling cost and improves accuracy at the same time; reducing labeling cost by 18.2%p and improving accuracy by 1.2%p at round 6. This result suggests that the proposed method can seamlessly incorporate advanced AL acquisition functions.

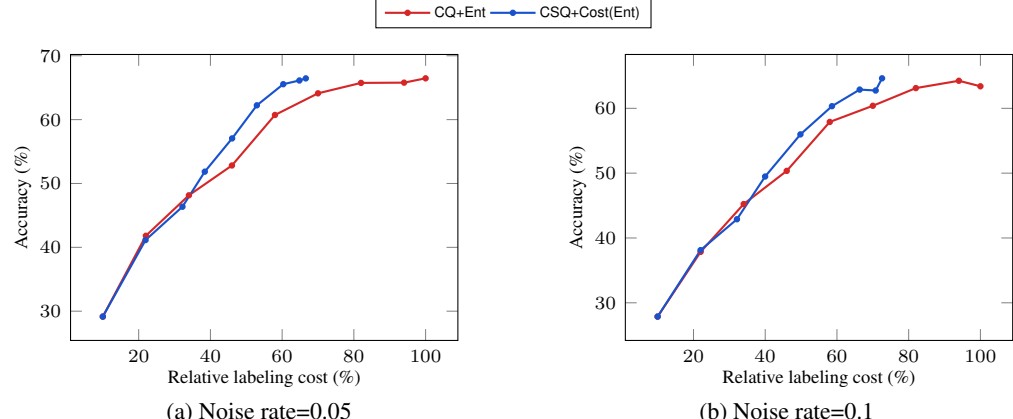

(a) Noise rate=0.05                    (b) Noise rate=0.1

Figure 15: Comparison between conventional query (CQ) and candidate set query (CSQ) with entropy sampling (Ent) and the proposed acquisition function with entropy measure (Cost(Ent)) on CIFAR-100 with label noise across AL rounds with varying noise level: (a) Noise rate of 0.05. (b) Noise rate of 0.1. The proposed CSQ+Cost(Ent) consistently outperforms CSQ+Ent across various AL rounds and noise rates.

## J  EXPERIMENTS ON REAL-WORLD DATASETS

**Experiment on datasets containing label noise.** We evaluate the candidate set query (CSQ) framework on CIFAR-100 with noisy labels, simulating a scenario where human annotators misclassify images into random classes with a noise rate $\epsilon$. This is modeled using a uniform label noise (Frénay & Verleysen, 2013) with $\epsilon$ set to 0.05 and 0.1. Note that this scenario is unfavorable for CSQ, as a misclassifying annotator would reject the actual true label even if the candidate set includes it.

Figure 15 compares CSQ and conventional query (CQ) on CIFAR-100 with noisy labels using entropy sampling (Ent) and our acquisition function with entropy measure (Cost(Ent)) across 2, 6, and 9 rounds.

Despite the disadvantageous scenario, our method (CSQ+Cost(Ent)) reduces labeling cost compared to the baseline (CQ+Ent) across varying AL rounds and noise rates. At round 9, CSQ+Cost(Ent) achieves cost reductions of 33.4%p and 27.4%p at noise rates of 0.05 and 0.1, respectively. It also consistently outperforms the baseline in terms of accuracy per labeling cost, demonstrating the robustness of CSQ.

Additionally, CSQ has the potential to reduce label noise, as narrowing the candidate set can lead to more precise annotations. Our user study (Table 1) shows that reducing candidate set size improves annotation accuracy, suggesting that CSQ can further enhance performance by reducing label noises.

**Experiment on datasets containing class imbalances.** Figure 16 compares candidate set query (CSQ) and conventional query (CQ) on CIFAR-100-LT (Cui et al., 2019), a class-imbalanced version of CIFAR-100, using entropy sampling (Ent), and our acquisition function with entropy measure (Cost(Ent)) across AL rounds. The experiments use imbalance ratios (*i.e.*, ratios between the largest and smallest class sizes) of 3, 6, and 10. Note that the maximum AL rounds vary with the imbalance ratio due to dataset size, with a maximum of 4 rounds for ratios of 3 and 6, and 6 rounds for a ratio of 10.

The result shows that our method (CSQ+Cost(Ent)) reduces labeling cost compared to the baselines (CQ+Ent) by significant margins across varying AL rounds and imbalance ratios. Specifically, at round 4, CSQ+Cost(Ent) achieves cost reductions of 31.1%p and 29.2%p at imbalance ratios of 6 and 10, respectively. In terms of accuracy per labeling cost, CSQ+Cost(Ent) consistently outperforms the baseline, demonstrating the robustness of the CSQ framework in class-imbalanced scenarios.

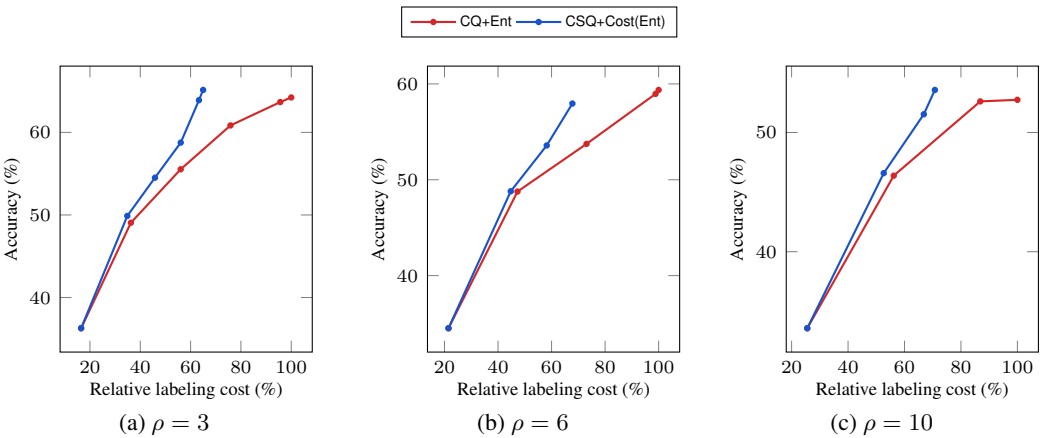

(a) $\rho = 3$        (b) $\rho = 6$        (c) $\rho = 10$

Figure 16: Comparison between conventional query (CQ) and candidate set query (CSQ) with entropy sampling (Ent) and the proposed acquisition function with entropy measure (Cost(Ent)) on CIFAR-100-LT, a version of CIFAR-100 with class imbalance, across AL rounds with varying imbalance level: (a) Imbalance ratio of 3. (b) Imbalance ratio of 6. (c) Imbalance ratio of 10. The proposed approach (CSQ+Cost(Ent)) consistently outperforms the baseline (CSQ+Ent) across various AL rounds and noise rates. Note that the maximum AL rounds vary with the imbalance ratio due to dataset size, with a maximum of 4 rounds for ratios of 3 and 6, and 6 rounds for a ratio of 10.

