# OpenReview forum: "Enhancing Cost Efficiency in Active Learning with Candidate Set Query"
_ICLR.cc/2025/Conference — Submitted to ICLR 2025_

### Official Review · Reviewer_QtBh · 2024-10-29

**Soundness:** 3
**Presentation:** 2
**Contribution:** 3
**Rating:** 6
**Confidence:** 3

**Summary:**

In this paper, the authors proposed a cost-efficient active learning (AL) framework for classification, featuring a novel query design called candidate set query (CSQ). CSQ narrows down the set of candidate classes likely to include the ground-truth class and leverages conformal prediction to dynamically generate small yet reliable candidate sets. Experiments on several datasets demonstrate the proposed CSQ is effective.

**Strengths:**

The proposed CSQ present the annotator with an image and a narrowed set of candidate classes that are likely to include the ground-truth class, which reduces labeling cost by minimizing the search space the annotator needs to explore. The various modules in the entire paradigm are relatively mature.

**Weaknesses:**

1. The explanation of "empirical quantile" in Section 3.2, titled "Candidate Set Construction from Conformal Prediction," is not very clear. Could the authors provide further clarification, specifically regarding the insights that Conformal Prediction offers for method design and its role in the proposed approach?

2. Is Table 1 in the experiments derived from the cost-efficient acquisition function?

3. Does the Active Learning (AL) approach include corresponding real-world datasets? Would it be possible to include more experimental results using real-world datasets?

**Questions:**

Please see the Weaknesses.

---

> ### Author Response · Authors · 2024-11-24
> **Response to Reviewer QtBh (Part 1/4)**
>
> ### **General reply**
> We thank you for your insightful review. We appreciate your suggestions, which have helped us improve our work. We have clarified our method below and hope our responses provide the necessary clarity. We also have revised the manuscript based on our responses, and all changes are highlighted in blue. If there are any additional questions, please do not hesitate to reach out.
>
> ---
>
> ### **Weakness 1. Clarification of empirical quantile in Section 3.2.**
> > The explanation of "empirical quantile" in Section 3.2, titled "Candidate Set Construction from Conformal Prediction," is not very clear. Could the authors provide further clarification, specifically regarding the insights that Conformal Prediction offers for method design and its role in the proposed approach?
>
> Thank you for your valuable comment, which helped us improve our manuscript.
>
> Conformal prediction allows candidate set query (CSQ) to reduce labeling costs by generating a candidate set that adjusts its size based on model knowledge.
>
> The empirical quantile $\hat{Q}(\alpha)$ in Eq. (4) of the paper serves as a threshold, defining the candidate set based on softmax values, as formulated in Eq. (5) of the paper. $\hat{Q}(\alpha)$ is estimated from the labeled calibration set $\mathcal{D}_\text{cal}$, and is expected to be generalized to the entire actively sampled dataset. This guarantees that the candidate set contains the correct label with a coverage of at least $100 \times (1-\alpha)$%, as shown in Eq. (6) of the paper. We have clarified Section 3.2, and added detailed procedures for calculating $\hat{Q}(\alpha)$ in Section C of the revised manuscript.
>
> If any parts remain unclear, please feel free to reach out with additional questions.
>
> ---
>
> ### **Weakness 2. Is Table 1 or the paper in the experiments derived from the cost-efficient acquisition function?**
>
> No.
> The images in the user study in Table 1 were randomly sampled from CIFAR-100 to avoid any sampling bias in the questionnaires.
> Further details about the user study are provided in Section A of the revised manuscript.

---

> ### Author Response · Authors · 2024-11-24
> **Response to Reviewer QtBh (Part 2/4)**
>
> ---
>
> ### **Weakness 3. More experimental results using real-world datasets**
> > Does the Active Learning (AL) approach include corresponding real-world datasets? Would it be possible to include more experimental results using real-world datasets?
>
> Thank you for the constructive comment.
>
> CIFAR-10, CIFAR-100, and ImageNet64x64, used in our paper, are well-known datasets for image recognition, including a diverse range of classes and scenes. Notably, we demonstrated the scalability of the candidate set query (CSQ) framework on ImageNet64x64, aligning with its goal of cost-efficient labeling for large-scale datasets. However, these datasets are primarily class-balanced, contain minimal label noise, and focus solely on the image recognition domain.
>
> To address your concerns, we conducted additional experiments on datasets containing CIFAR-100 with label noise and imbalanced classes, and experiments in a natural language domain (text classification).
>
> **Experiment on datasets containing imbalance classes**
>
> Table R7 compares candidate set query (CSQ) and conventional query (CQ) on CIFAR-100-LT (Cui et al., 2019), a class-imbalanced version of CIFAR-100, using entropy sampling (Ent), and our acquisition function with entropy measure (Cost(Ent)) across 2, 3, and 4 rounds. The experiments use imbalance ratios of 3, 6, and 10, defined as the ratio between the largest and smallest class sizes. Results for all active learning (AL) rounds are provided in Figure 16 of the revised manuscript. Note that the maximum AL rounds vary with the imbalance ratio due to dataset size, with a maximum of 4 rounds for ratios of 3 and 6, and 6 rounds for a ratio of 10.
>
> The results show that our method (CSQ+Cost(Ent)) reduces labeling cost compared to the baselines (CQ+Ent) by significant margins across varying AL rounds and imbalance ratio. Specifically, at round 4, CSQ+Cost(Ent) achieves cost reductions of 32.1%p and 29.2%p at imbalance ratios of 6 and 10, respectively. In terms of accuracy per labeling cost, CSQ+Cost(Ent) consistently outperforms the baseline, demonstrating the robustness of CSQ framework in class-imbalanced scenarios.
>
> #### **Table R7**: Relative labeling cost (\%) and accuracy (\%) of conventional query (CQ) and our candidate set query (CSQ) on CIFAR-100-LT (Cao et al., 2021) with Entropy (Ent) and our acquisition function with entropy measure (Cost(Ent), Eq. (8)). Imbalance ratio indicates the ratio between the largest and smallest class sizes.
>
> | Method | AL round | Imbalance ratio | Relative Labeling Cost (\%) | Accuracy (\%) | Accuracy per Cost |
> | :--- | :---: | :---: | :---: | :---: | :---: |
> | CQ+Ent | 2 | 3 | 36.3 | 49.1 | 1.35 |
> | CSQ+Cost(Ent) | 2 | 3 | **34.8 (-1.5\%p)** | 49.9 | **1.43** |
> | CQ+Ent | 2 | 6 | 47.3 | 48.8 | 1.03 |
> | CSQ+Cost(Ent) | 2 | 6 | **44.8 (-2.5\%p)** | 48.8 | **1.09** |
> | CQ+Ent | 2 | 10 | 56.2 | 46.4 | 0.83 |
> | CSQ+Cost(Ent) | 2 | 10 | **52.6 (-3.6\%p)** | 46.6 | **0.89** |
> | ------------------- | --------------- | --------------- | ---------------------------------------- | --------------------- | --------------------- |
> | CQ+Ent | 3 | 3 | 56.1 | 55.5 | 0.99 |
> | CSQ+Cost(Ent) | 3 | 3 | **45.8 (-10.3\%p)** | 54.5 | **1.19** |
> | CQ+Ent | 3 | 6 | 73.1 | 53.7 | 0.73 |
> | CSQ+Cost(Ent) | 3 | 6 | **58.2 (-14.9\%p)** | 53.6 | **0.92** |
> | CQ+Ent | 3 | 10 | 86.9 | 52.6 | 0.61 |
> | CSQ+Cost(Ent) | 3 | 10 | **66.8 (-20.1\%p)** | 51.5 | **0.77** |
> | ------------------- | --------------- | --------------- | ---------------------------------------- | --------------------- | --------------------- |
> | CQ+Ent | 4 | 3 | 75.8 | 60.8 | 0.80 |
> | CSQ+Cost(Ent) | 4 | 3 | **56.1 (-19.7\%p)** | 58.8 | **1.05** |
> | CQ+Ent | 4 | 6 | 98.9 | 59.0 | 0.60 |
> | CSQ+Cost(Ent) | 4 | 6 | **67.8 (-31.1\%p)** | 58.0 | **0.86** |
> | CQ+Ent | 4 | 10 | 100.0 | 52.7 | 0.53 |
> | CSQ+Cost(Ent) | 4 | 10 | **70.8 (-29.2\%p)** | 53.6 | **0.76** |
>
> ##### *Relative Labeling Cost (\%): Smaller is better
>
> We will continue our discussion for this concern in the following comment section, due to the space limit.

---

> ### Author Response · Authors · 2024-11-24
> **Response to Reviewer QtBh (Part 3/4)**
>
> **Experiment on datasets containing label noise**
>
> We evaluate the candidate set query (CSQ) framework on CIFAR-100 with noisy labels, simulating a scenario where human annotators misclassify images into random classes with a noise rate $\epsilon$. This is modeled using a uniform label noise (Frénay and Verleysen, 2013) with $\epsilon$ set to 0.05 and 0.1. Note that this scenario is unfavorable for CSQ, as a misclassifying annotator would reject the actual truth label even if the candidate set includes it.
>
> Table R8 compares CSQ and conventional query (CQ) on CIFAR-100 with noisy labels using entropy sampling (Ent) and our acquisition function with entropy measure (Cost(Ent)) across 2, 6, and 9 rounds. Results for all active learning (AL) rounds are provided in Figure 15 of the revised manuscript.
>
> Despite the disadvantageous scenario, our method (CSQ+Cost(Ent)) reduces labeling cost compared to the baseline (CQ+Ent) across varying AL rounds and noise rates. At round 9, CSQ+Cost(Ent) achieves cost reductions of 33.4%p and 27.4%p at noise rates of 0.05 and 0.1, respectively. It also consistently outperforms the baseline in terms of accuracy per labeling cost, demonstrating the robustness of CSQ.
>
> Additionally, CSQ has the potential to reduce label noise, as narrowing the candidate set can lead to more precise annotations. Our user study (Table 1 of the paper) shows that reducing candidate set size improves annotation accuracy, suggesting that CSQ can further enhance performance by reducing label noises.
>
> #### **Table R8**: Relative labeling cost (\%) and accuracy (\%) of conventional query (CQ) and our candidate set query (CSQ) on CIFAR-100 with noisy labels with Entropy (Ent) and our acquisition function with entropy measure (Cost(Ent), Eq. (8) of the paper). The noise rate indicates the proportion of incorrect annotations.
>
> | Method | AL round | Noise Rates | Relative Labeling Cost (\%) | Accuracy (\%) | Accuracy per Cost |
> | :--- | :---: | :---: | :---: | :---: | :---: |
> | CQ+Ent | 2 | 0.05 | 22.0 | 41.2 | 1.87 |
> | CSQ+Cost(Ent) | 2 | 0.05 | **21.9 (-0.1\%)** | 41.1 | **1.88** |
> | CQ+Ent | 2 | 0.1 | 22.0 | 37.9 | 1.72 |
> | CSQ+Cost(Ent) | 2 | 0.1 | **21.9 (-0.1\%)** | 38.1 | **1.74** |
> | ------------------- | --------------- | --------------- | ---------------------------------------- | --------------------- | --------------------- |
> | CQ+Ent | 6 | 0.05 | 70.0 | 64.1 | 0.92 |
> | CSQ+Cost(Ent) | 6 | 0.05 | **53.0 (-17\%p)** | 62.2 | **1.17** |
> | CQ+Ent | 6 | 0.1 | 70.0 | 60.4 | 0.86 |
> | CSQ+Cost(Ent) | 6 | 0.1 | **58.6 (-11.4\%p)** | 60.3 | **1.03** |
> | ------------------- | --------------- | --------------- | ---------------------------------------- | --------------------- | --------------------- |
> | CQ+Ent | 9 | 0.05 | 100.0 | 66.5 | 0.67 |
> | CSQ+Cost(Ent) | 9 | 0.05 | **66.6 (-33.4\%p)** | 66.5 | **1.00** |
> | CQ+Ent | 9 | 0.1 | 100.0 | 63.4 | 0.63 |
> | CSQ+Cost(Ent) | 9 | 0.1 | **72.6 (-27.4\%p)** | 64.6 | **0.89** |
>
> ##### *Relative Labeling Cost (\%): Smaller is better
>
> We will continue our discussion for this concern in the following comment section, due to the space limit.

---

> ### Author Response · Authors · 2024-11-24
> **Response to Reviewer QtBh (Part 4/4)**
>
> **Experiment on Text classification**
>
> Table R9 present a comparison of candidate set query (CSQ) and conventional query (CQ) on text classification dataset (R52, Lewis, 1997) with random sampling (Rand), entropy sampling (Ent), and our acquisition function with entropy measure (Cost(Ent)) across 4, 7, and 10 rounds. Results across all active learning (AL) rounds are shown in Figure 13 of the revised manuscript. CSQ approaches consistently outperform the CQ baselines by a significant margin across various budgets and acquisition functions. Especially at round 10, CSQ+Rand reduces labeling cost by 65.6\%p compared to its conventional query baseline. The result demonstrates that the proposed CSQ framework generalizes to the text classification domain.
>
> #### **Table R9**: Relative labeling cost (\%) and accuracy (\%) of conventional query (CQ) and our candidate set query (CSQ) on text classification dataset (R52) with Entropy (Ent) and cost-efficient sampling (Cost(Ent), Eq. (8)). Since candidate set design does not impact sampling, the same acquisition function yields identical accuracy.
>
> | Method | AL round | Relative Labeling Cost (\%) | Accuracy (\%) |
> | :--- | :---: | :---: | :---: |
> | CQ+Rand | 4 | 35.7 | 89.5 |
> | CSQ+Rand | 4 | **16.3 (-9.4\%)** | 89.5 |
> | CQ+Ent | 4 | 35.7 | 90.8 |
> | CSQ+Cost(Ent) | 4 | **16.4 (-19.3\%)** | 90.8 |
> | ------------------- | --------------- | ---------------------------------------- | --------------------- |
> | CQ+Rand | 7 | 66.3 | 92.6 |
> | CSQ+Rand | 7 | **24.3 (-44.0\%)** | 92.6 |
> | CQ+Ent | 7 | 66.3 | 93.1 |
> | CSQ+Cost(Ent) | 7 | **26.3 (-40.0\%)** | 93.1 |
> | ------------------- | --------------- | ---------------------------------------- | --------------------- |
> | CQ+Rand | 10 | 97.0 | 93.5 |
> | CSQ+Rand | 10 | **31.4 (-65.6\%)** | 93.5 |
> | CQ+Ent | 10 | 97.0 | 93.9 |
> | CSQ+Cost(Ent) | 10 | **34.5 (-62.5\%)** | 93.8 |
>
> ##### *Relative Labeling Cost (\%): Smaller is better
>
> These results and more detailed configuration about the experiment have been added to Section J (class imbalance and label noise) and Section H (text classification) of the revised manuscript.
>
> ---
> ---
>
> ### **References**
>
> Lewis, D. D. (1997). Reuters-21578 text categorization test collection.
>
> Du, P., Zhao, S., Chen, H., Chai, S., Chen, H., & Li, C. (2021). Contrastive coding for active learning under class distribution mismatch. In Proceedings of the IEEE/CVF International Conference on Computer Vision (pp. 8927-8936).
>
> Wei, J., Zhu, Z., Cheng, H., Liu, T., Niu, G., & Liu, Y. (2022). Learning with noisy labels revisited: A study using real-world human annotations. Proc. International Conference on Learning Representations (ICLR).
>
> Cui, Y., Jia, M., Lin, T. Y., Song, Y., & Belongie, S. (2019). Class-balanced loss based on effective number of samples. In Proceedings of the IEEE/CVF conference on computer vision and pattern recognition (pp. 9268-9277).
>
> Cao, K., Wei, C., Gaidon, A., Arechiga, N., & Ma, T. (2019). Learning imbalanced datasets with label-distribution-aware margin loss. Advances in neural information processing systems, 32.
>
> Cui, Y., Jia, M., Lin, T. Y., Song, Y., & Belongie, S. (2019). Class-balanced loss based on effective number of samples. In Proceedings of the IEEE/CVF conference on computer vision and pattern recognition (pp. 9268-9277).
>
> Frénay, B., & Verleysen, M. (2013). Classification in the presence of label noise: a survey. IEEE transactions on neural networks and learning systems, 25(5), 845-869.

---

> ### Author Response · Authors · 2024-11-29
> **Request for feedback on the revised manuscript**
>
> We sincerely appreciate the time and effort you dedicated to reviewing our work. Your comments have significantly contributed to improving its quality.
>
> In the revised manuscript, we have enhanced the clarity of the empirical quantile through a detailed explanation in Section C, clarified the specifics of the user study, and added experiments on datasets reflecting realistic scenarios.
>
> Furthermore, we have expanded the appendix to include additional experiments on a text classification task, an alternative acquisition function, a hyperparameter analysis, and an ablation study focused on low-confidence samples.
>
> We would be grateful if you could review the updated manuscript and share your valuable feedback. Your suggestions would play a crucial role in further refining our work.

---

### Official Review · Reviewer_sJcA · 2024-10-31

**Soundness:** 3
**Presentation:** 3
**Contribution:** 3
**Rating:** 6
**Confidence:** 4

**Summary:**

This paper introduces the Candidate Set Query (CSQ) framework which aims to improve the cost efficiency in active learning (AL) tasks. CSQ reduces the size of the set of candidate classes which reduces the search space, leveraging conformal prediction to dynamically adjust the optimal candidate set size across successive AL rounds. The paper also proposes an acquisition function that considers the ratio of information gain vs labeling cost to help prioritize samples to label. The author then benchmarked this framework across multiple datasets and empirically demonstrate the reduction in labeling costs.

**Strengths:**

1. The motivation for this paper is clear, and the paper proposes a novel framework of high significance
2. The paper presents a solid theoretical framework that is thoroughly explained and mostly straightforward to follow
3. The framework is benchmarked across 3 well-known datasets, empirically demonstrating the effectiveness of the method
4. Thorough ablations studies were conducted to highlight the significance of each component of the framework

**Weaknesses:**

1. The benchmarks are conducted on very similar datasets (CIFAR-10, CIFAR-100, and ImageNet64x64 are all image classification datasets), and also only compares against a small number of baseline AL methods. It is unclear if the results will generalize well across different datasets and domains, and if more advanced underlying AL acquisition methods are used
2. The paper does not consider the implication of real-world datasets, such as those containing label noise, imbalance classes etc might impact the performance of CSQ. More experiments could help identify CSQ’s robustness when it comes to noisy annotations, as it could lead to inefficient training and candidate sets with lower quality
3. CSQ relies on several hyperparameters, however there is limited justification on how to properly optimize the hyperparameter $d$, making it difficult to apply the method in new datasets

**Questions:**

1. Have the authors considered any special handling of outlier or anomalous datapoints, which tend to have have high uncertainty, and could lead to inefficient construction of the candidate set, acquisition and labeling
2. Have the authors considered the risk of overfitting and drift of confidence scores across successive AL rounds and how CSQ could incorporate some steps to address it?

---

> ### Author Response · Authors · 2024-11-24
> **Response to Reviewer sJcA (Part 1/5)**
>
> ### **General reply**
> Thank you for your detailed and encouraging feedback. We appreciate your thoughtful evaluation of our contributions and inspiring comments. We have provided additional experiments and responses below to address your concerns. We also have revised the manuscript based on our responses, and all changes are highlighted in blue. We hope our clarifications meet your expectations, and we remain available to answer any further questions.

---

> ### Author Response · Authors · 2024-11-24
> **Response to Reviewer sJcA (Part 2/5)**
>
> ### **Weakness 1. Generalization to different domain and AL acquisition methods**
> > The benchmarks are conducted on very similar datasets (CIFAR-10, CIFAR-100, and ImageNet64x64 are all image classification datasets), and also only compares against a small number of baseline AL methods. It is unclear if the results will generalize well across different datasets and domains, and if more advanced underlying AL acquisition methods are used.
>
> Thank you for the constructive comment. To address your concerns, we conducted additional experiments in a different domain (text classification) and with a recently proposed active learning acquisition method, ProbCover (Yehuda et al., 2022).
>
> **Experiment on Text classification**
>
> Table R3 presents a comparison of candidate set query (CSQ) and conventional query (CQ) on a text classification dataset (R52, Lewis, 1997) with random sampling (Rand), entropy sampling (Ent), and our acquisition function with entropy measure (Cost(Ent), Eq. (8) of the paper) across 4, 7, and 10 rounds. CSQ  consistently outperforms the CQ baselines by a significant margin across various budgets and acquisition functions. Especially at round 10, CSQ+Rand reduces labeling cost by 65.6\%p compared to its CQ counterpart. This result suggests that the proposed CSQ framework well generalizes to text classification.
>
> #### **Table R3**: Relative labeling cost (\%) and accuracy (\%) of conventional query (CQ) and our candidate set query (CSQ) on text classification dataset (R52) with Entropy (Ent) and cost-efficient sampling (Cost(Ent), Eq. (8) of the paper). As candidate set design does not impact sampling, the same acquisition function yields identical accuracy.
>
> | Method | AL round | Relative Labeling Cost (\%) | Accuracy (\%) |
> | :--- | :---: | :---: | :---: |
> | CQ+Rand | 4 | 35.7 | 89.5 |
> | CSQ+Rand | 4 | **16.3 (-9.4\%)** | 89.5 |
> | CQ+Ent | 4 | 35.7 | 90.8 |
> | CSQ+Cost(Ent) | 4 | **16.4 (-19.3\%)** | 90.8 |
> | ---------------------- | -------------- | --------------------------------------- | -------------------- |
> | CQ+Rand | 7 | 66.3 | 92.6 |
> | CSQ+Rand | 7 | **24.3 (-44.0\%)** | 92.6 |
> | CQ+Ent | 7 | 66.3 | 93.1 |
> | CSQ+Cost(Ent) | 7 | **26.3 (-40.0\%)** | 93.1 |
> | ---------------------- | -------------- | --------------------------------------- | -------------------- |
> | CQ+Rand | 10 | 97.0 | 93.5 |
> | CSQ+Rand | 10 | **31.4 (-65.6\%)** | 93.5 |
> | CQ+Ent | 10 | 97.0 | 93.9 |
> | CSQ+Cost(Ent) | 10 | **34.5 (-62.5\%)** | 93.8 |
>
> ##### *Relative Labeling Cost (\%): Smaller is better
>
> Results across all active learning (AL) rounds are shown in Figure 13 of the revised manuscript. More details of the dataset and implementation about the text classification experiment is provided in Section H of the revised manuscript.
>
> **Experiment with ProbCover acquisition function**
>
> First, we would like to clarify that the candidate set query design is orthogonal to the sampling method, as it improves annotation efficiency independently of the acquisition function. This means it can be combined with any acquisition function. Further, our cost-efficient design of acquisition function (Eq. (8) of the paper) can be applied to any existing acquisition functions.
>
> To show this versatility empirically, and to demonstrate the generalization ability of our acquisition function, in Table R4, we compare CSQ and CQ on CIFAR-100 with ProbCover (Yehuda et al., 2022) sampling and cost-efficient ProbCover sampling (Cost(ProbCover)), across 4, 7, and 10 rounds. Results across all active learning (AL) rounds are shown in Figure 14 of the revised manuscript. CSQ consistently outperforms the CQ baselines across various budgets and acquisition functions. In particular, the proposed method reduces labeling cost and improves accuracy at the same time; reducing labeling cost by 18.2\%p and improving accuracy by 1.2\%p at round 6. This result suggests that the proposed method can seamlessly incorporate advanced AL acquisition functions.
>
> #### Table R4: Relative labeling cost (\%) and accuracy (\%) of conventional query (CQ) and our candidate set query (CSQ) on CIFAR-100 paired with ProbCover (Yehuda et al., 2022) and our acquisition function ProbCover measure (Cost(ProbCover)).
>
> | Method | AL round | Relative Labeling Cost (\%) | Accuracy (\%) |
> | :--- | :---: | :---: | :---: |
> | CQ+ProbCover | 3 | 34.0 | 51.5 |
> | CSQ+Cost(ProbCover) | 3 | **29.2 (-4.8\%)** | 52.9 |
> | -------------------------------- | --------------- | --------------------------------------- | -------------------- |
> | CQ+ProbCover | 6 | 70.0 | 68.5 |
> | CSQ+Cost(ProbCover) | 6 | **50.3 (-18.2\%)** | 69.7 |
> | -------------------------------- | --------------- | --------------------------------------- | -------------------- |
> | CQ+ProbCover | 9 | 100.0 | 71.8 |
> | CSQ+Cost(ProbCover) | 9 | **57.2 (-14.6\%)** | 72.5 |
>
> ##### *Relative Labeling Cost (\%): Smaller is better
>
> These results of CSQ paired with ProbCover have been added to Section I of the revised manuscript.

---

> ### Author Response · Authors · 2024-11-24
> **Response to Reviewer sJcA (Part 3/5)**
>
> ### **Weakness 2. Robustness to real-world datasets**
> > The paper does not consider the implication of real-world datasets, such as those containing label noise, imbalance classes etc might impact the performance of CSQ. More experiments could help identify CSQ’s robustness when it comes to noisy annotations, as it could lead to inefficient training and candidate sets with lower quality.
>
> Thank you for the comment. To address your concerns, we conducted additional experiments on datasets containing CIFAR-100 with label noise and imbalanced classes.
>
> **Experiment on datasets containing label noise**
>
> We evaluate the candidate set query (CSQ) framework on CIFAR-100 with noisy labels, simulating a scenario where human annotators misclassify images into random classes with a noise rate $\epsilon$. This is modeled using a uniform label noise (Frénay and Verleysen, 2013) with $\epsilon$ set to 0.05 and 0.1. Note that this scenario is unfavorable for CSQ, as a misclassifying annotator would reject the actual truth label even if the candidate set includes it.
>
> Table R5 compares CSQ and conventional query (CQ) on CIFAR-100 with noisy labels using entropy sampling (Ent) and our acquisition function with entropy measure (Cost(Ent)) across 2, 6, and 9 rounds. Results for all active learning (AL) rounds are provided in Figure 15 of the revised manuscript.
>
> Despite the disadvantageous scenario, our method (CSQ+Cost(Ent)) reduces labeling cost compared to the baseline (CQ+Ent) across varying AL rounds and noise rates. At round 9, CSQ+Cost(Ent) achieves cost reductions of 33.4%p and 27.4%p at noise rates of 0.05 and 0.1, respectively. It also consistently outperforms the baseline in terms of accuracy per labeling cost, demonstrating the robustness of CSQ.
>
> Additionally, CSQ has the potential to reduce label noise, as narrowing the candidate set can lead to more precise annotations. Our user study (Table 1 of the paper) shows that reducing candidate set size improves annotation accuracy, suggesting that CSQ can further enhance performance by reducing label noises.
>
> #### **Table R5**: Relative labeling cost (\%) and accuracy (\%) of conventional query (CQ) and our candidate set query (CSQ) on CIFAR-100 with noisy labels with Entropy (Ent) and our acquisition function with entropy measure (Cost(Ent), Eq. (8) of the paper). The noise rate indicates the proportion of incorrect annotations.
>
> | Method | AL round | Noise Rates | Relative Labeling Cost (\%) | Accuracy (\%) | Accuracy per Cost |
> | :--- | :---: | :---: | :---: | :---: | :---: |
> | CQ+Ent | 2 | 0.05 | 22.0 | 41.2 | 1.87 |
> | CSQ+Cost(Ent) | 2 | 0.05 | **21.9 (-0.1\%)** | 41.1 | **1.88** |
> | CQ+Ent | 2 | 0.1 | 22.0 | 37.9 | 1.72 |
> | CSQ+Cost(Ent) | 2 | 0.1 | **21.9 (-0.1\%)** | 38.1 | **1.74** |
> | ------------------- | --------------- | --------------- | ---------------------------------------- | --------------------- | --------------------- |
> | CQ+Ent | 6 | 0.05 | 70.0 | 64.1 | 0.92 |
> | CSQ+Cost(Ent) | 6 | 0.05 | **53.0 (-17\%p)** | 62.2 | **1.17** |
> | CQ+Ent | 6 | 0.1 | 70.0 | 60.4 | 0.86 |
> | CSQ+Cost(Ent) | 6 | 0.1 | **58.6 (-11.4\%p)** | 60.3 | **1.03** |
> | ------------------- | --------------- | --------------- | ---------------------------------------- | --------------------- | --------------------- |
> | CQ+Ent | 9 | 0.05 | 100.0 | 66.5 | 0.67 |
> | CSQ+Cost(Ent) | 9 | 0.05 | **66.6 (-33.4\%p)** | 66.5 | **1.00** |
> | CQ+Ent | 9 | 0.1 | 100.0 | 63.4 | 0.63 |
> | CSQ+Cost(Ent) | 9 | 0.1 | **72.6 (-27.4\%p)** | 64.6 | **0.89** |
>
> ##### *Relative Labeling Cost (\%): Smaller is better
>
> We will continue our discussion for this concern in the following comment section, due to the space limit.

---

> ### Author Response · Authors · 2024-11-24
> **Response to Reviewer sJcA (Part 4/5)**
>
> **Experiments on datasets with class imbalance**
>
> Table R6 compares candidate set query (CSQ) and conventional query (CQ) on CIFAR-100-LT (Cui et al., 2019), a class-imbalanced version of CIFAR-100, using entropy sampling (Ent), and our acquisition function with entropy measure (Cost(Ent)) across 2, 3, and 4 rounds. The experiments use imbalance ratios (i.e., ratios between the largest and smallest class sizes) of 3, 6, and 10. Results for all active learning (AL) rounds are provided in Figure 16 of the revised manuscript. Note that the maximum AL rounds vary with the imbalance ratio due to dataset size, with a maximum of 4 rounds for ratios of 3 and 6, and 6 rounds for a ratio of 10.
>
> The result shows that our method (CSQ+Cost(Ent)) reduces labeling cost compared to the baselines (CQ+Ent) by significant margins across varying AL rounds and imbalance ratios. Specifically, at round 4, CSQ+Cost(Ent) achieves cost reductions of 31.1%p and 29.2%p at imbalance ratios of 6 and 10, respectively. In terms of accuracy per labeling cost, CSQ+Cost(Ent) consistently outperforms the baseline, demonstrating the robustness of the CSQ framework in class-imbalanced scenarios.
>
> #### **Table R6**: Relative labeling cost (\%) and accuracy (\%) of conventional query (CQ) and our candidate set query (CSQ) on CIFAR-100-LT (Cao et al., 2021) with Entropy (Ent) and our acquisition function with entropy measure (Cost(Ent), Eq. (8)). Imbalance ratio indicates the ratio between the largest and smallest class sizes.
>
> | Method | AL round | Imbalance ratio | Relative Labeling Cost (\%) | Accuracy (\%) | Accuracy per Cost |
> | :--- | :---: | :---: | :---: | :---: | :---: |
> | CQ+Ent | 2 | 3 | 36.3 | 49.1 | 1.35 |
> | CSQ+Cost(Ent) | 2 | 3 | **34.8 (-1.5\%p)** | 49.9 | **1.43** |
> | CQ+Ent | 2 | 6 | 47.3 | 48.8 | 1.03 |
> | CSQ+Cost(Ent) | 2 | 6 | **44.8 (-2.5\%p)** | 48.8 | **1.09** |
> | CQ+Ent | 2 | 10 | 56.2 | 46.4 | 0.83 |
> | CSQ+Cost(Ent) | 2 | 10 | **52.6 (-3.6\%p)** | 46.6 | **0.89** |
> | ------------------- | --------------- | --------------- | ---------------------------------------- | --------------------- | --------------------- |
> | CQ+Ent | 3 | 3 | 56.1 | 55.5 | 0.99 |
> | CSQ+Cost(Ent) | 3 | 3 | **45.8 (-10.3\%p)** | 54.5 | **1.19** |
> | CQ+Ent | 3 | 6 | 73.1 | 53.7 | 0.73 |
> | CSQ+Cost(Ent) | 3 | 6 | **58.2 (-14.9\%p)** | 53.6 | **0.92** |
> | CQ+Ent | 3 | 10 | 86.9 | 52.6 | 0.61 |
> | CSQ+Cost(Ent) | 3 | 10 | **66.8 (-20.1\%p)** | 51.5 | **0.77** |
> | ------------------- | --------------- | --------------- | ---------------------------------------- | --------------------- | --------------------- |
> | CQ+Ent | 4 | 3 | 75.8 | 60.8 | 0.80 |
> | CSQ+Cost(Ent) | 4 | 3 | **56.1 (-19.7\%p)** | 58.8 | **1.05** |
> | CQ+Ent | 4 | 6 | 98.9 | 59.0 | 0.60 |
> | CSQ+Cost(Ent) | 4 | 6 | **67.8 (-31.1\%p)** | 58.0 | **0.86** |
> | CQ+Ent | 4 | 10 | 100.0 | 52.7 | 0.53 |
> | CSQ+Cost(Ent) | 4 | 10 | **70.8 (-29.2\%p)** | 53.6 | **0.76** |
>
> ##### *Relative Labeling Cost (\%): Smaller is better
>
> These results on realistic scenarios of datasets with label noise and class imbalances have been added to Section J of the revised manuscript, respectively.
>
> ---
>
> ### **Weakness 3. Guideline for selecting hyperparameter $d$ in new dataset**
> > CSQ relies on several hyperparameters, however there is limited justification on how to properly optimize the hyperparameter $d$, making it difficult to apply the method in new datasets.
>
> Thank you for the helpful comment on practical application. We provide an analysis showing the trend of both labeling cost and accuracy with varying $d$ values over AL rounds for CIFAR-10, CIFAR-100, and ImageNet64x64 in Figure 10.
>
> In CIFAR-10 (Figure 10a), both accuracy and labeling cost remain robust to the change of $d$, varying only 0.5\%p in accuracy and 2.0\%p in labeling cost. In CIFAR-100 (Figure 10b), the overall performance seems to increase as $d$ decreases. On the other hand, in ImageNet64x64 (Figure 10c), the performance decreases as $d$ increases until it reaches 2.0. This aligns with recent observations that uncertainty-based selection performs better in scenarios with larger labeling budgets (Hacohen et al., 2022), as increasing $d$ prioritizes uncertain samples.
>
> Based on these results, we provide the following guidelines for setting $d$:
>
> - For datasets with fewer than 100 classes, d values between 0.3 and 1.0 may be effective, as they ensure robustness on simpler datasets like CIFAR-10 and reduce labeling costs on more complex datasets like CIFAR-100.
> - For larger datasets closer in scale to ImageNet, exploring $d \geq 1.0$ can help further improve the model performance.
>
> This analysis and the proposed guidelines have been added to Section E of the revised manuscript.

---

> ### Author Response · Authors · 2024-11-24
> **Response to Reviewer sJcA (Part 5/5)**
>
> ### **Question 1. Handling of outlier or anomalous data points.**
> > Have the authors considered any special handling of outlier or anomalous data points, which tend to have high uncertainty, and could lead to inefficient construction of the candidate set, acquisition and labeling.
>
> Thank you for the insightful question.
>
> Dealing with out-of-distribution (OOD) data points showing high uncertainty scores has been a chronic issue in active learning and may affect the efficiency of candidate set query (CSQ). Recent open-set active learning approaches (Du et al., 2021; Kothawade et al., 2021; Ning et al., 2022; Park et al., 2022; Yang et al., 2024) tackle this by filtering out OOD samples during active sampling using an OOD classifier. Our CSQ framework integrates seamlessly with these methods, focusing on labeling in-distribution (ID) samples to prevent cost inefficiencies.
>
> However, as OOD classifiers are not flawless, some OOD samples may still be selected. One advantage of our method is its ability to leverage the calibration set to capture information about such mixed OOD samples. This enables adjustments such as increasing the OOD classifier threshold to exclude more OOD-like data or incorporating the OOD ratio into the alpha optimization process in Eq. (7) of the paper.
>
> Optimizing the combination of OOD and ID classifier scores within the calibration set or designing better OOD-aware queries presents promising future research directions.
>
> We have added a discussion about issues on OOD sample in Section D of the revised manuscript, and included references to open-set active learning literature.
>
>
> ---
>
> ### **Question 2. Consideration on the risk of overfitting and drift of confidence scores**
> > Have the authors considered the risk of overfitting and drift of confidence scores across successive AL rounds and how CSQ could incorporate some steps to address it?
>
> The risk of overfitting or drift in confidence scores is minimal since these scores are derived from a calibration dataset that has not been used during the model training phase. This ensures that the candidate set query maintains a theoretical guarantee of including the correct class with a predefined probability (Eq. (6) in the paper), regardless of the model or data distribution (Vovk et al., 1999; Angelopoulos et al., 2023). We have clarified the condition of the calibration set in Section 3.2 of the revised manuscript.
>
>
> ---
> ---
>
>
> ### **References**
>
> Yehuda, O., Dekel, A., Hacohen, G., & Weinshall, D. (2022). Active learning through a covering lens. *Advances in Neural Information Processing Systems*, *35*, 22354-22367.
>
> Lewis, D. D. (1997). Reuters-21578 text categorization test collection.
>
> Du, P., Zhao, S., Chen, H., Chai, S., Chen, H., & Li, C. (2021). Contrastive coding for active learning under class distribution mismatch. In Proceedings of the IEEE/CVF International Conference on Computer Vision (pp. 8927-8936).
>
> Kothawade, S., Beck, N., Killamsetty, K., & Iyer, R. (2021). Similar: Submodular information measures based active learning in realistic scenarios. Advances in Neural Information Processing Systems, 34, 18685-18697.
>
> Ning, K. P., Zhao, X., Li, Y., & Huang, S. J. (2022). Active learning for open-set annotation. In Proceedings of the IEEE/CVF Conference on Computer Vision and Pattern Recognition (pp. 41-49).
>
> Park, D., Shin, Y., Bang, J., Lee, Y., Song, H., & Lee, J. G. (2022). Meta-query-net: Resolving purity-informativeness dilemma in open-set active learning. Advances in Neural Information Processing Systems, 35, 31416-31429.
>
> Yang, Y., Zhang, Y., Song, X., & Xu, Y. (2024). Not all out-of-distribution data are harmful to open-set active learning. Advances in Neural Information Processing Systems, 36.
>
> Wei, J., Zhu, Z., Cheng, H., Liu, T., Niu, G., & Liu, Y. (2022). Learning with noisy labels revisited: A study using real-world human annotations. Proc. International Conference on Learning Representations (ICLR).
>
> Cao, K., Wei, C., Gaidon, A., Arechiga, N., & Ma, T. (2019). Learning imbalanced datasets with label-distribution-aware margin loss. Advances in neural information processing systems, 32.
>
> Frénay, B., & Verleysen, M. (2013). Classification in the presence of label noise: a survey. IEEE transactions on neural networks and learning systems, 25(5), 845-869.

---

> > ### Comment · Reviewer_sJcA · 2024-11-27
> >
> > Thank you for the detailed response and the additional details. The responses satisfy most of my concerns, and I appreciate seeing more experiments conducted that demonstrate that this method is generalizable across different modalities, and is robust towards real-world datasets. I would rate to accept this paper.

---

> > > ### Author Response · Authors · 2024-11-27
> > >
> > > Thank you for your quick response and for the positive feedback! We are happy to hear that most of your concerns have been addressed.
> > >
> > > If there are any remaining concerns that make you hesitant to increase your score from 6 (marginally above the acceptance threshold), please let us know. We are happy to respond!

---

### Official Review · Reviewer_YK4X · 2024-11-04

**Soundness:** 2
**Presentation:** 3
**Contribution:** 2
**Rating:** 5
**Confidence:** 4

**Summary:**

The paper proposes a cost-efficient active learning strategy using conformal predictions. Instead of letting the annotator choose from all possible labels, a candidate set of fewer labels is given. The paper uses a log order expected cost function and shows the improved efficiency in terms of actual labeling cost.

**Strengths:**

1. The content of the paper is well presented.
2. The paper studies the cost of AL query in a more realistic way and proposes a solution for reducing the cost by candidate set query.
3. The candidate set is formed by conformal prediction and the candidate labels are related to the expected information gain with cost considerations.

**Weaknesses:**

The proposed method still depends on the conformal prediction and the calibration set to determine the confidence level. It is a realistic solution however not guaranteed to be theoretically sound. The convergence can not be obtained in a proper label complexity analysis. Similarly, the labeling cost assumption in Theorem 3.1 is only a rough approximation.

**Questions:**

1. Is random selection effective for the calibration set?
2. Is it possible that the candidate set obtained from eq(5) is empty?
3. Is there any guarantee that using the quantile from the previous round would work?

---

> ### Author Response · Authors · 2024-11-20
> **Response to Reviewer YK4X (Part 1/3)**
>
> ### **General reply**
> We greatly appreciate your constructive feedback. Please find below our responses to your comments and questions. We have done our best to understand your intentions, but if there are any aspects that still need further elaboration, please feel free to let us know. We also have revised the manuscript based on our responses, and all changes are highlighted in blue.
>
> ---
>
> ### **Weakness 1. The proposed method still depends on the conformal prediction and the calibration set to determine the confidence level.**
>
> We would like to first clarify the definition and role of the confidence level. The confidence level (line 136 of the paper), i.e., $1-\alpha$ in Eq. (6), is an essential hyperparameter for conformal prediction, reflecting user preference of covering the correct label; users who want the candidate set more likely to contain the true label will set $\alpha$ to a smaller value. Hence, it does not depend on the calibration set in the conformal prediction literature.
>
> The comment is partially valid in that we present a novel strategy to dynamically adjust the confidence level ($1-\alpha^*$ in Eq. (7)) using the calibration set (Section 3.2). We consider this as a strength rather than a weakness since it removes the hyperparameter $\alpha$ and adapts to the model over successive active learning rounds, thereby improving the performance. Figure 6 demonstrates that this strategy consistently enhances labeling cost efficiency.
>
> In addition, the use of the calibration set does not affect the performance of the trained model, and has very little impact on the cost efficiency of the entire active learning process, as it is obtained within the allocated budget and also contributes to model training. The performance of CSQ remains robust to varying calibration set sizes, as shown in Figure 4b.
>
> We clarified the use of the calibration set and confidence level optimization at line 239 and line 266 in Section 3 of the revised manuscript.
>
> ---
>
> ### **Weakness 2. It is a realistic solution however not guaranteed to be theoretically sound. The convergence cannot be obtained in a proper label complexity analysis.**
>
>
> We appreciate the valuable feedback regarding the limitations in theoretical guarantees about label complexity.
>
> Our main contribution, the candidate set query (CSQ) design, is orthogonal to the sampling method, as it improves annotation efficiency independently of the acquisition function. This means it can be combined with any acquisition function, including theoretically rigorous ones. Figures 2 and 4 show that CSQ reduces labeling costs across image classification datasets with various acquisition functions.
>
> Unfortunately, the proposed cost-efficient acquisition function lacks theoretical guarantee for label complexity at this point due to its reliance on heuristic informativeness measures (both entropy and BADGE) and the inherent complexity of deep learning models. In the same context, to the best of our knowledge, recent deep active learning methods lack theoretical analysis on label complexity and focus primarily on empirical validation (Asghar et al., 2017; He et al., 2019; Ostapuk et al., 2019; Fuchsgruber et al., 2024; Sener & Savarese, 2018; Sinha et al., 2019; Yehuda et al., 2022; Ash et al., 2020; Hwang et al., 2022; Wang & Ye, 2015; Kim et al., 2024; Cho et al., 2024). Our method shows practical effectiveness in reducing labeling costs, as shown in Figure 4a.
>
> We agree that rigorous theoretical guarantees for label complexity will be a valuable contribution to the community. While this remains an open challenge, we believe our work contributes to the active learning literature in the following two ways:
>
> 1. An efficient annotation query design that can be seamlessly integrated with various active learning strategies
> 2. Cost-aware acquisition function that accounts for instance-specific labeling costs
>
> We have updated the limitation and future work about label complexity analysis at line 533 in Section 5 of the revised manuscript.

---

> ### Author Response · Authors · 2024-11-20
> **Response to Reviewer YK4X (Part 2/3)**
>
> ### **Weakness 3. The labeling cost assumption in Theorem 3.1 is only a rough approximation.**
>
> First of all, we would like to stress that Theorem 3.1 holds for any cost models increasing monotonically with the number of labeling options (line 215 of the paper), and that such cost models hold for general annotation processes (i.e., assigning tags to individual samples).
>
> The labeling cost model in Theorem 3.1 is based on the principle that an accurate label provides $\log_2 C$ bits of information (Hu et al., 2020), where $C$ is the number of classes.
> This cost model is empirically supported by the user studies conducted both in our work and in prior research (Hu et al., 2020).
>
> Figure 1(*right*) and Table 1 of the paper present the results of our user study with 40 participants, demonstrating that the empirical labeling cost increases logarithmically to the number of possible options.
> This finding aligns closely with the theoretical cost model, showing a correlation coefficient of 0.97.
> Details of our user study are provided in Appendix A.
>
> ---
>
> ### **Question 1. Is random selection effective for the calibration set?**
>
> Yes.
> The calibration set must be randomly sampled from the actively selected data to satisfy the
> requirement of conformal prediction (Vovk et al., 1999; Angelopoulos et al., 2023), ensuring coverage in Eq. (6).
> To achieve this, we first select $B$ samples using the acquisition function and then randomly sample the calibration set from these $B$ samples, where $B$ is the labeling budget.
> In Figure 5c, Conformal($\alpha=0.1$) demonstrates that with $\alpha$ of 0.1, the candidate set includes the correct label in over 90% of cases, empirically verifying the coverage in Eq. (6).
>
> We have clarified the conformal prediction requirement for the calibration set at line 235 in Section 3 of the revised manuscript.
>
> ---
>
> ### **Question 2. Is it possible that the candidate set obtained from eq(5) is empty?**
>
> Yes.
> It is possible for a candidate set to be empty if the calibration set happens to consist of samples with highly confident predictions on the ground truth class, and a new sample has a nearly uniform, flat predicted probability.
> However, such corner cases are extremely rare, occurring in only about 0.024% of cases when training on CIFAR-100 with CSQ+Cost(Ent) in Figure 2.
> We note that these corner cases do not negatively affect the efficiency of CSQ or the coverage ensured by Eq. (6).
> For simplicity, we implemented our method to include at least the top-1 predicted class.
>
> ---
>
> ### **Question 3. Is there any guarantee that using the quantile from the previous round would work?**
>
> No.
> To ensure the coverage in Eq. (6), the quantile needs to be recalculated from the calibration data in the current round.
> This is why our method recalculates the quantile in Eq. (4) each round, allowing the candidate set query to guarantee coverage within the active learning framework.
>
> On the other hand, when labeling the calibration set, the quantile in the current round is not available.
> Thus, we use candidate set query without a coverage guarantee for these calibration samples using the quantile from the previous round to maximize the efficiency of our method. Approximately 8% of the budget in each round is allocated to the calibration set.

---

> ### Author Response · Authors · 2024-11-20
> **Response to Reviewer YK4X (Part 3/3)**
>
> ### **References**
>
> Hu, H., Xie, L., Du, Z., Hong, R., & Tian, Q. (2020). One-bit supervision for image classification. Advances in Neural Information Processing Systems, 33, 501-511.
>
> Vovk, V., Gammerman, A., & Saunders, C. (1999). Machine-learning applications of algorithmic randomness.
>
> Angelopoulos, A. N., & Bates, S. (2023). Conformal prediction: A gentle introduction. Foundations and Trends® in Machine Learning, 16(4), 494-591.
>
> Asghar, N., Poupart, P., Jiang, X., & Li, H. (2017, August). Deep Active Learning for Dialogue Generation. In N. Ide, A. Herbelot, & L. Màrquez (Eds.), Proceedings of the 6th Joint Conference on Lexical and Computational Semantics (*SEM 2017) (pp. 78–83).
>
> He, T., Jin, X., Ding, G., Yi, L., & Yan, C. (2019). Towards Better Uncertainty Sampling: Active Learning with Multiple Views for Deep Convolutional Neural Network. 2019 IEEE International Conference on Multimedia and Expo (ICME), 1360–1365.
>
> Ostapuk, N., Yang, J., & Cudré-Mauroux, P. (2019). Activelink: deep active learning for link prediction in knowledge graphs. The World Wide Web Conference, 1398–1408.
>
> Fuchsgruber, D., Wollschläger, T., Charpentier, B., Oroz, A., & Günnemann, S. (2024). Uncertainty for Active Learning on Graphs. Proc. International Conference on Learning Representations (ICLR).
>
> Sener, O., & Savarese, S. (2018). Active Learning for Convolutional Neural Networks: A Core-Set Approach. International Conference on Learning Representations.
>
> Sinha, S., Ebrahimi, S., & Darrell, T. (2019). Variational adversarial active learning. Proceedings of the IEEE/CVF International Conference on Computer Vision, 5972–5981.
>
> Yehuda, O., Dekel, A., Hacohen, G., & Weinshall, D. (2022). Active learning through a covering lens. Advances in Neural Information Processing Systems, 35, 22354–22367.
>
> Ash, J. T., Zhang, C., Krishnamurthy, A., Langford, J., & Agarwal, A. (2020). Deep Batch Active Learning by Diverse, Uncertain Gradient Lower Bounds. International Conference on Learning Representations.
>
> Hwang, S., Lee, S., Kim, S., Ok, J., & Kwak, S. (2022). Combating label distribution shift for active domain adaptation. European Conference on Computer Vision, 549–566. Springer.
>
> Wang, Z., & Ye, J. (2015). Querying discriminative and representative samples for batch mode active learning. ACM Transactions on Knowledge Discovery from Data (TKDD), 9(3), 1–23.
>
> Kim, H., Hwang, S., Kwak, S., & Ok, J. (2024). Active Label Correction for Semantic Segmentation with Foundation Models. Proc. International Conference on Machine Learning (ICML).
>
> Cho, S. J., Kim, G., Lee, J., Shin, J., & Yoo, C. D. (2024). Querying Easily Flip-flopped Samples for Deep Active Learning. Proc. International Conference on Learning Representations (ICLR).

---

> > ### Comment · Reviewer_YK4X · 2024-11-27
> >
> > Thank you for the response. Some of my questions are addressed. However, the concern over the convergence guarantee and using the previous round still remains. I also agree with Reviewer 6wYp on the uncertainty quantification comment. The problem is multi-class and as long as we stick with probabilities we are restricted to first-order uncertainties. It is probable that the concluded method is still found in a set of sub-optimal solutions and selected based on heuristics.

---

> ### Author Response · Authors · 2024-11-29
> **Further Response to Reviewer YK4X (Part 1/2)**
>
> Thank you for your feedback. We appreciate the time and effort you have dedicated to reviewing our work. We are glad to hear that some of your concerns, including our confidence level settings, cost model, and calibration set sampling, have been addressed.
>
> As far as we understand, the remaining concerns are the utilization of the quantile of previous rounds when labeling the calibration set, uncertainty quantification, and the convergence guarantee. Below, we provide detailed responses to these concerns.
>
> ---
>
> ### **Using quantile from previous round for calibration set labeling**
>
>
> > However, the concern over the convergence guarantee and using the previous round still remains.
>
> We would like to clarify that the proposed method primarily utilizes the quantile values of *the current round*. The quantile values computed in the previous round are used only for labeling the calibration set, and, moreover, the set does not necessarily need to be labeled in that way. For example, the calibration set can be labeled with the conventional query (i.e., presenting all possible options as candidate classes), and this approach incurs only a tiny additional cost as the calibration set takes up only 8\% of the budget in each round, while completely eliminating dependency on the previous round. Nonetheless, our method references the quantile values from the previous round since it does not introduce additional computation or resource overhead while helping save labeling costs as much as possible.
>
> Table R10 demonstrates the performance of candidate set query (CSQ) when the calibration set is labeled with the conventional query without dependency on the previous round. Even without this dependency, our method significantly reduces cost compared to the baseline method (CQ+Ent).
>
> #### **Table R10**: Relative labeling cost (%) and accuracy (%) of conventional query (CQ), candidate set query (CSQ), and CSQ with using CQ for labeling calibration set, evaluated on CIFAR-100 with entropy sampling (Ent) and the proposed acquisition function with entropy measure (Cost(Ent)).
> | Method | Usage of prev. round | Round | Relative Labeling Cost (\%) | Acc. (\%) | Acc. per Cost |
> | :--- | :---: | :---: | :---: | :---: | :---: |
> | CQ + Ent | No | 3 | 34.0 | 55.1 | 1.62 |
> | CSQ + Cost(Ent) | Yes | 3 | 29.4 (-4.6\%p) | 53.3 | 1.81 |
> | CSQ + Cost(Ent) *CQ for $\mathcal{D}_\text{cal}$ | No | 3 | 29.8 (-4.2\%p) | 53.3 | 1.79 |
> | ------------------------------------- | ---------------------------- | --------- | ------------------------------------ | ------------ | ---------------- |
> | CQ + Ent | No | 6 | 70.0 | 70.2 | 1.00 |
> | CSQ + Cost(Ent) | Yes | 6 | 49.1 (-20.9\%p) | 66.7 | 1.36 |
> | CSQ + Cost(Ent) *CQ for $\mathcal{D}_\text{cal}$ | No | 6 | 50.9 (-19.1\%p) | 66.7 | 1.31 |
> | ------------------------------------- | ---------------------------- | --------- | ------------------------------------ | ------------ | ---------------- |
> | CQ + Ent | No | 9 | 100.0 | 72.3 | 0.72 |
> | CSQ + Cost(Ent) | Yes | 9 | 62.1 (-37.9\%p) | 72.2 | 1.16 |
> | CSQ + Cost(Ent) *CQ for $\mathcal{D}_\text{cal}$ | No | 9 | 65.6 (-34.4\%p) | 72.2 | 1.10 |
> ##### *Relative Labeling Cost (\%): Smaller is better
>
> ---
>
> ### **Uncertainty quantification**
>
> > I also agree with Reviewer 6wYp on the uncertainty quantification comment. The problem is multi-class and as long as we stick with probabilities we are restricted to first-order uncertainties. It is probable that the concluded method is still found in a set of sub-optimal solutions and selected based on heuristics.
>
> We also agree that Reviewer 6wYp provided an insightful comment and experimental suggestion regarding the case of softmax values that are close to uniform, and a comparison with a baseline sifting out classes with low softmax values. To address this concern, we have demonstrated in Figure 12 of the revised manuscript that CSQ performs effectively even under entropy-based acquisition, which favors uniform softmax values. In Figure 11 of the revised manuscript, we also demonstrated that CSQ outperforms the sifting baseline (CSQ-sift) suggested by Reviewer 6wYp.
>
> Meanwhile, we note that uncertainty quantification (i.e., nonconformity measure) in conformal prediction is not restricted to softmax values (Vovk, 2005; Angelopoulos et al., 2023); alternative methods of uncertainty quantification or additional adjustments to the softmax values can also be employed (Angelopoulos et al., 2020; Ding et al., 2023; Fong & Holmes, 2021). We choose softmax values as it is simple and known to be effective with deep learning models (Romano et al., 2020), which enables us to reduce labeling cost effectively on various benchmarks.
>
> Exploring more advanced nonconformity measures or identifying ones particularly effective for the proposed candidate set query framework could bring further improvements, which we leave as future work.

---

> ### Author Response · Authors · 2024-11-29
> **Further Response to Reviewer YK4X (Part 2/2)**
>
> ### **Convergence guarantee**
>
> > However, the concern over the convergence guarantee and using the previous round still remains.
>
> We stress that candidate set query (CSQ) design is orthogonal to the sampling method, and enhances annotation efficiency independently of the choice of acquisition function. *Therefore, it can also be combined with acquisition functions that have convergence guarantees*. Figures 2 and 4 demonstrate that CSQ reduces labeling costs across various acquisition functions. Additionally, as shown in Figure 14 of the revised manuscript, we have newly validated the effectiveness of CSQ when combined with ProbCover (Yehuda et al., 2022).
>
> We acknowledge that the proposed acquisition function currently lacks a convergence guarantee due to its reliance on heuristic informativeness measures, such as entropy and BADGE, and the inherent complexity of deep learning models. *This limitation is common among most of the deep active learning methods*. Nevertheless, we believe that, even without a convergence guarantee, the extensive empirical validation of such methods has made significant contributions to the literature. In particular, the effectiveness of our work has been validated across a variety of domains and realistic scenarios, including: large-scale datasets (ImageNet64x64, Figure 2c of the paper), noisy-labeled datasets (CIFAR-100 with label noise; Frénay and Verleysen, 2013, Figure 15 of the revised manuscript), datasets with class imbalances (CIFAR-100-LT; Cui et al., 2019, Figure 16 of the revised manuscript), and text classification (R52; Lewis, 1997, Figure 13 of the revised manuscript).
>
> ---
>
> We hope our clarifications meet your expectations, and we are happy to address any further questions or concerns you may have.
>
> ---
> ---
>
> ### **References**
>
> Yehuda, O., Dekel, A., Hacohen, G., & Weinshall, D. (2022). Active learning through a covering lens. Advances in Neural Information Processing Systems, 35, 22354–22367.
>
> Angelopoulos, A. N., & Bates, S. (2023). Conformal prediction: A gentle introduction. Foundations and Trends® in Machine Learning, 16(4), 494-591.
>
> Fong, E., & Holmes, C. C. (2021). Conformal bayesian computation. Advances in Neural Information Processing Systems, 34, 18268-18279.
>
> Cao, K., Wei, C., Gaidon, A., Arechiga, N., & Ma, T. (2019). Learning imbalanced datasets with label-distribution-aware margin loss. Advances in neural information processing systems, 32.
>
> Frénay, B., & Verleysen, M. (2013). Classification in the presence of label noise: a survey. IEEE transactions on neural networks and learning systems, 25(5), 845-869.
>
> Lewis, D. D. (1997). Reuters-21578 text categorization test collection.
>
> Vovk, V., Gammerman, A., & Shafer, G. (2005). Algorithmic learning in a random world (Vol. 29). New York: Springer.
>
> Romano, Y., Sesia, M., & Candes, E. (2020). Classification with valid and adaptive coverage. Advances in Neural Information Processing Systems, 33, 3581-3591.
>
> Angelopoulos, A., Bates, S., Malik, J., & Jordan, M. I. (2020). Uncertainty sets for image classifiers using conformal prediction. arXiv preprint arXiv:2009.14193.
>
> Ding, T., Angelopoulos, A., Bates, S., Jordan, M., & Tibshirani, R. J. (2023). Class-conditional conformal prediction with many classes. Advances in Neural Information Processing Systems.

---

### Official Review · Reviewer_6wYp · 2024-11-04

**Soundness:** 3
**Presentation:** 3
**Contribution:** 2
**Rating:** 5
**Confidence:** 4

**Summary:**

This paper proposes an active-learning framework called Candidate Set Query (CSQ) to reduce the size of possible classes during annotation process, minimizing the search space and lowering the labeling cost at the same time. Furthermore, the authors leverage conformal prediction to produce accurate candidate sets, and introduce an acquisition function that exploits data points with high information gain.

**Strengths:**

1. This paper introduces a novel approach called Candidate Set Query (CSQ), which effectively reduces labeling costs by narrowing down the candidate classes presented to annotators, thereby minimizing annotation time.
2. The proposed method leverages conformal prediction to dynamically produce accurate candidate labels based on a cost-efficient data acquisition function. This function prioritizes samples with high information gain, leading to greater efficiency and reduced labeling costs.
3. The framework demonstrates strong performance across multiple image recognition datasets, consistently outperforming baseline methods by a significant margin.
4. The authors provide comprehensive ablation studies to thoroughly validate the effectiveness of all components.

**Weaknesses:**

1. The rationale behind the cost-efficient acquisition function in Eq. (8) needs to be further explained. Additional motivation and explanation for this function are recommended.
2. As shown in Fig. 9a, the performance is sensitive to the hyperparameter d. Providing guidelines for setting this parameter to an appropriate range on different datasets would be beneficial.
3. In realistic scenarios, the samples with high uncertainty waiting to be annotated can be divided into two groups based on their probability distributions on categories, high confidence on several specific classes or low confidence on almost all classes. The proposed framework might only be suitable for the former case. As for the latter one, an intuitive solution is to sift out all candidate labels with low prediction probabilities, such as less than 0.1 * 1/C where C is the number of categories. So it’s suggested to conduct more experiments to evaluate the performance of this approach under this scenario.
4. A smaller candidate label set means model have higher certainty for samples, which seems to be against the motivation of selecting the most uncertain samples in active learning. Despite the proposed method minimizing the labeling cost by narrowing down the label space, the information gain is limited. It’s suggested to plot the graph of how performance varies with the number of queried samples.
5. There is a minor formatting issue in Fig. 5c, where certain data points from the top-1 prediction fall outside the scale range of the graph.
6. Line 274 contains a typo: “calcuclate” should be corrected to “calculate.”

**Questions:**

Please refer to the above weaknesses.

---

> ### Author Response · Authors · 2024-11-24
> **Response to Reviewer 6wYp (Part 1/4)**
>
> ### **General reply**
> We sincerely appreciate your thorough review and valuable feedback. Below we provide additional explanations and experiments to address your concerns. The manuscript has been modified accordingly, and all changes are highlighted in blue. If there are any remaining concerns, please let us know, and we will gladly provide further clarification.
>
> ---
>
> ### **Weakness 1. The rationale behind the cost-efficient acquisition function in Eq. (8)**
> > The rationale behind the cost-efficient acquisition function in Eq. (8) needs to be further explained. Additional motivation and explanation for this function are recommended.
>
> The main motivation behind the cost-efficient acquisition function is that, given the same budget, a large number of inexpensive and (slightly) less informative samples can be more effective than a few expensive and most informative samples in CSQ. Considering this, our acquisition function is designed to measure information gain per unit cost (Line 282 of the manuscript). The denominator is the estimated labeling cost of a sample, while the numerator is its estimated information gain.
>
> Specifically, the estimated labeling cost of a sample $\mathbf{x}$ in the denominator is expressed as:
>
> $$
> (1-\alpha^*)  \text{log}_2 (k+1) + \alpha^* (\text{log}_2 (k+1) + \text{log}_2 (L-k)),
> $$
>
> where $k := |\hat{Y}_{\theta}(\mathbf{x}, \alpha^*)|$ is the size of the candidate set (Eq. (5) of the paper), and $L$ is the number of classes. The equation is the expected cost derived from our cost model (Eq. (1) of the paper), accounting for both cases: when the correct label is included in the candidate set (first term) and when it is excluded (second term). This expected cost assumes the probability of the candidate set missing the correct label as error rate $\alpha^*$ (Eq. (7) of the paper), which is supported by the fact that the candidate set theoretically guarantees to cover the correct class with probability greater than $1-\alpha^*$ (Eq. (6) of the paper, Vovk et al., 1999; Angelopoulos et al., 2023).
>
> Meanwhile, the estimated information gain in the numerator is quantified using acquisition scores from existing active learning (AL) methods, as these are already well-established. For a given sample $\mathbf{x}$, this is defined as:
>
> $$
> (1 + g_{\text{score}}(\mathbf{x}))^d,
> $$
>
> where $g_{\text{score}}(\mathbf{x})$ is the acquisition score from an existing method, and $d$ is a hyperparameter. As any existing acquisition functions with different ranges can be employed for $g_{\text{score}}(\mathbf{x})$, we normalize $g_{\text{score}}(\mathbf{x})$ so that its range should be $[0, 1]$. We apply the exponent $d$ to adjust the balance between the information gain and labeling cost, with 1 added to the base to give intuitive scaling as $d$ increases.
>
> Figure 4a shows that our acquisition function consistently improves cost-efficiency. We revised Section 3.3 of the manuscript to include these details.
>
> ---
>
> ### **Weakness 2. Guidelines for setting hyperparameter $d$**
> > As shown in Fig. 9a, the performance is sensitive to the hyperparameter d. Providing guidelines for setting this parameter to an appropriate range on different datasets would be beneficial.
>
> Thank you for the helpful suggestion.
>
>
> First of all, we would like to clarify that Figure 9 shows how the hyperparameter $d$ affects labeling cost, not the performance of trained models; the labeling cost sensitive to $d$ is natural and desirable as we intentionally designed it that way.
>
> To address your concern, we provide a more comprehensive analysis showing the trend of performance in accuracy with varying $d$ values over AL rounds for CIFAR-10, CIFAR-100, and ImageNet64x64 in Figure 10. In CIFAR-10 (Figure 10a), both the accuracy and labeling cost remain robust to the change of $d$, varying only 0.5\%p in accuracy. In CIFAR-100 (Figure 10b), the overall performance is still insensitive yet slightly increasing as $d$ decreases. On the other hand, in ImageNet64x64 (Figure 10c), the performance decreases as $d$ increases until it reaches 2.0. Regarding that a larger $d$ prioritizes more uncertain samples, this result aligns with recent observations that uncertainty-based selection performs better in scenarios with larger labeling budgets (Hacohen et al., 2022).
>
> Based on these results, we provide the following guidelines for setting $d$:
>
> - For datasets with fewer than 100 classes, $d$ values between 0.3 and 1.0 may be effective, as they ensure robustness on simple datasets like CIFAR-10 and reduce labeling costs on more complex datasets like CIFAR-100.
> - For larger datasets closer in scale to ImageNet, exploring $d \geq 1.0$ can help further improve the model performance.
>
> This analysis and the proposed guidelines have been added to Section E of the revised manuscript.

---

> ### Author Response · Authors · 2024-11-24
> **Response to Reviewer 6wYp (Part 2/4)**
>
> ### **Weakness 3. Potential limitation of the proposed framework in handling low-confidence samples**
> > In realistic scenarios, the samples with high uncertainty waiting to be annotated can be divided into two groups based on their probability distributions on categories, high confidence on several specific classes or low confidence on almost all classes. The proposed framework might only be suitable for the former case. As for the latter one, an intuitive solution is to sift out all candidate labels with low prediction probabilities, such as less than 0.1 * 1/C where C is the number of categories. So it’s suggested to conduct more experiments to evaluate the performance of this approach under this scenario.
>
> We sincerely thank the reviewer for the constructive comment.
>
> We would like to first stress that sifting out candidate labels with low prediction probabilities is essentially equivalent to a heuristic variant of CSQ, where the threshold for the prediction probability $1 - \hat{Q}(\alpha)$ (Eq. (5) of the paper) is fixed manually. In contrast, CSQ adaptively adjusts the threshold to minimize the labeling cost by the error rate optimization (Eq. (7) of the paper).
>
> For the scenario of low confidence on almost all classes, entropy-based sampling is particularly relevant, as entropy explicitly reflects the uniformity of a probability distribution. To demonstrate the effectiveness of CSQ under this scenario, Table R1 compares CSQ with the aforementioned variant, dubbed CSQ-sift, that sifts out classes with softmax values below $0.1 \times 1/C$ ($C$: number of classes), using entropy and BADGE (Ash et al., 2020) sampling on CIFAR-100 across the active learning (AL) rounds 3, 6, and 9.
>
> The results show that CSQ is more cost-efficient, reducing relative labeling cost by 7.2%p compared to CSQ-sift at round 9 even with entropy sampling, favoring samples with uniform softmax values. When paired with BADGE, a more advanced diversity-aware acquisition function, CSQ shows additional cost savings. Results for all active learning (AL) rounds are provided in Figure 11 of the revised manuscript.
>
> #### **Table R1**: Relative labeling cost (\%) and accuracy (\%) of conventional query (CQ), candidate set query (CSQ) and sifting-based baseline (CSQ-sift) on CIFAR-100 with entropy and BADGE (Ash et al., 2020) sampling. Since candidate set design does not impact sampling, the same acquisition function yields identical accuracy.
>
> | Method | AL round | Relative Labeling Cost (\%) | Accuracy (\%) |
> | :--- | :--- | :---: | :---: |
> | CQ + Ent | 3 | 34.0 | 55.1 |
> | CSQ-sift + Ent  | 3 | 32.6 | 55.1 |
> | CSQ + Ent | 3 | **32.2 (-0.4\%p)** | 55.1 |
> | ------------------- | --------------- | ---------------------------------------- | --------------------- |
> | CQ + Ent | 6 | 70.0 | 70.2 |
> | CSQ-sift + Ent  | 6 | 60.0 | 70.2 |
> | CSQ + Ent | 6 | **54.4 (-5.6\%p)** | 70.2 |
> | ------------------- | --------------- | ---------------------------------------- | --------------------- |
> | CQ + Ent | 9 | 100.0 | 72.3 |
> | CSQ-sift + Ent  | 9 | 68.0 | 72.3 |
> | CSQ + Ent | 9 | **60.8 (-7.2\%p)** | 72.3 |
> ##### *Relative Labeling Cost (\%): Smaller is better
>
> | Method | AL round | Relative Labeling Cost (\%) | Accuracy (\%) |
> | :--- | :--- | :---: | :---: |
> | CQ + BADGE | 3 | 34.0 | 55.9 |
> | CSQ-sift + BADGE  | 3 | 27.7 | 55.9 |
> | CSQ + BADGE | 3 | **25.5 (-2.2\%p)** | 55.9 |
> | ------------------------- | --------------- | ---------------------------------------- | --------------------- |
> | CQ + BADGE | 6 | 70.0 | 68.3 |
> | CSQ-sift + BADGE  | 6 | 49.8 | 68.3 |
> | CSQ + BADGE | 6 | **43.1 (-6.7\%p)** | 68.3 |
> | ------------------------- | --------------- | ---------------------------------------- | --------------------- |
> | CQ + BADGE | 9 | 100.0 | 72.1 |
> | CSQ-sift + BADGE  | 9 | 63.7 | 72.1 |
> | CSQ + BADGE | 9 | **57.4 (-6.3\%p)** | 72.1 |
> ##### *Relative Labeling Cost (\%): Smaller is better
>
> It is worth mentioning that situations where the model exhibits low confidence across almost all classes are rare, primarily due to the inherent overconfidence of deep learning models (Guo et al., 2017). For example, at AL round 6, even for the sample with the highest entropy, the top-10 of softmax values account for approximately 62\% of the total probability.
>
> CSQ also offers a key advantage over the heuristic variant (CSQ-sift) by providing a theoretical guarantee of including the correct class (Eq. (6) of the paper), enabling the use of our acquisition function (Eq. (8) of the paper). This acquisition function further enhances cost-efficiency, as shown in Figure 4a of the paper.
>
> We have added a discussion and experimental results regarding CSQ-sift in Section F of the revised manuscript.

---

> ### Author Response · Authors · 2024-11-24
> **Response to Reviewer 6wYp (Part 3/4)**
>
> ### **Weakness 4. Potential conflict between uncertainty-based selection and narrowed label space in the proposed method**
> > A smaller candidate label set means model have higher certainty for samples, which seems to be against the motivation of selecting the most uncertain samples in active learning. Despite the proposed method minimizing the labeling cost by narrowing down the label space, the information gain is limited. It’s suggested to plot the graph of how performance varies with the number of queried samples.
>
> We would like to stress that our framework focuses on maximizing the total information gain of selected data by increasing the quantity of the data within a fixed cost. While traditional uncertainty-based sampling prioritizes high per-sample information, recent advancements like BADGE (Ash et al., 2020) emphasize maximizing the total information gain by increasing the diversity of selected samples as well as the uncertainty. Aligning with this trend, our method increases total information gain by increasing the number of labeled samples while reducing labeling cost per sample.
>
> We achieve this through two key components: query design and acquisition function. To address your concerns, we show that (1) our acquisition function outperforms uncertainty-based acquisition function in accuracy per cost, and (2) our query design reduces labeling costs even for highly uncertain samples.
>
> Table R2 compares CSQ and conventional query (CQ) on CIFAR-100 with entropy-based sampling (Ent) and our acquisition function with entropy measure (Cost(Ent), Eq.(8) of th paper) across 3, 6, and 9 rounds, with a fixed number of samples per round. Results for all AL rounds are shown in Figure 12 of the revised manuscript.
>
> **Our acquisition function provides superior accuracy per cost.** The comparison between CSQ+Cost(Ent) and CSQ+Ent demonstrates that the proposed acquisition function reduces labeling costs with only a marginal accuracy trade-off. When considering accuracy per labeling cost, our acquisition function consistently outperforms the entropy baseline, which is also well illustrated in Figure 12a.
>
> **Candidate set query (CSQ) can reduce labeling costs even for uncertain samples.** The comparison between CQ+Ent and CSQ+Ent demonstrates that CSQ effectively reduces labeling costs, even with uncertainty-based sampling methods like entropy sampling. This shows that CSQ can narrow down annotation options even for uncertain samples. Note that CSQ+Ent shows the same accuracy as CQ+Ent, since they used the same sampling method.
>
> #### **Table R2**: Relative labeling cost (\%) and accuracy (\%) of conventional query (CQ) and our candidate set query (CSQ) on CIFAR-100 with Entropy (Ent) and our acquisition function with entropy measure (Cost(Ent), Eq.(8) of the paper). Since query design does not impact sampling, the same acquisition function yields identical accuracy.
>
> | Method | AL round | # labeled samples | Relative Labeling Cost (\%) | Accuracy (\%) | Accuracy per Cost |
> | :--- | :---: | :---: | :---: | :---: | :---: |
> | CQ + Ent | 3 | 17000 | 34.0 | 55.1 | 1.62 |
> | CSQ + Ent | 3 | 17000 | 32.2 | 55.1 | 1.71 |
> | CSQ + Cost(Ent) | 3 | 17000 | **27.8 (-4.4\%)** | 54.3 | **1.95** |
> | ------------------- | --------------- | --------------- | ---------------------------------------- | --------------------- | --------------------- |
> | CQ + Ent | 6 | 35000 | 70.0 | 70.2 | 1.00 |
> | CSQ + Ent | 6 | 35000 | 54.4 | 70.2 | 1.29 |
> | CSQ + Cost(Ent) | 6 | 35000 | **46.5 (-7.9\%)** | 68.9 | **1.48** |
> | ------------------- | --------------- | --------------- | ---------------------------------------- | --------------------- | --------------------- |
> | CQ + Ent | 9 | 50000 | 100.0 | 72.3 | 0.72 |
> | CSQ + Ent | 9 | 50000 | 60.8 | 72.3 | 1.19 |
> | CSQ + Cost(Ent) | 9 | 50000 | **56.8 (-4.0\%)** | 72.9 | **1.28** |
>
>
> ##### *Relative Labeling Cost (\%): Smaller is better
>
> We incorporated these discussions and results into Section G of the revised manuscript.
>
> ---
>
> ### **Weakness 5.  A minor formatting issue in Fig. 5c.**
> > There is a minor formatting issue in Fig. 5c, where certain data points from the top-1 prediction fall outside the scale range of the graph.
>
> Thank you for your thorough review. We have updated and corrected Figure 5c in the revised manuscript.
>
> ---
>
> ### **Weakness 6.  A typo in Line 274**
> > Line 274 contains a typo: “calcuclate” should be corrected to “calculate.”
>
> Thank you for pointing this out. We have corrected the typo in the revised manuscript.

---

> ### Author Response · Authors · 2024-11-24
> **Response to Reviewer 6wYp (Part 4/4)**
>
> ### **References**
>
> Vovk, V., Gammerman, A., & Saunders, C. (1999). Machine-learning applications of algorithmic randomness.
>
> Angelopoulos, A. N., & Bates, S. (2023). Conformal prediction: A gentle introduction. Foundations and Trends® in Machine Learning, 16(4), 494-591.
>
> Guy Hacohen, Avihu Dekel, and Daphna Weinshall.(2022). Active learning on a budget: Opposite strategies suit high and low budgets. International Conference on Machine Learning (pp. 8175–8195). PMLR.
>
> Ash, J. T., Zhang, C., Krishnamurthy, A., Langford, J., & Agarwal, A. (2020). Deep Batch Active Learning by Diverse, Uncertain Gradient Lower Bounds. International Conference on Learning Representations.
>
> Guo, C., Pleiss, G., Sun, Y., & Weinberger, K. Q. (2017, July). On calibration of modern neural networks. In International conference on machine learning (pp. 1321-1330). PMLR.

---

> ### Author Response · Authors · 2024-11-29
> **Request for feedback on the revised manuscript**
>
> We deeply appreciate the time and effort you dedicated to reviewing our work. Your insightful comments have helped us make meaningful improvements.
>
> In the revised version, we have clarified the details of the acquisition function, analyzed the impact of the hyperparameter $d$, and conducted an in-depth evaluation of low-confidence samples, including a comparison with the sift-out baseline you suggested.
>
> Additionally, we expanded the appendix to include experiments on new datasets simulating realistic scenarios, a text classification task, and additional acquisition functions.
>
> We would be grateful if you could take the time to share your feedback on the revised version, as your suggestions would greatly contribute to refining our work further.

---

### Author Response · Authors · 2024-11-20
**General Response to Reviewers**

We sincerely thank the reviewers for their valuable feedback and are encouraged by their recognition of the novelty (**6wYp**, **sJcA**) and soundness (**6wYp**, **YK4X**, **sJcA**, **QtBh**) of our work. We also appreciate their positive comments on our use of conformal prediction (**6wYp**, **YK4X**), the strong empirical performance of our method (**6wYp**, **sJcA**), and the thoroughness of our ablation studies (**6wYp**, **sJcA**).

We address each reviewer's comments and questions on each thread. All modifications made to the manuscript are highlighted in blue.

---

### Author Response · Authors · 2024-12-04
**Summary of reviewers’ overall comments and the corresponding responses**

We thank the four reviewers and the Area Chair for their time and effort in reviewing our work and managing the review process. For your convenience, we have summarized the overall review comments and our responses during the rebuttal period below.

### **Summary of strength recognized by reviewers**

We are encouraged by the reviewers recognizing that our method is a novel (**6wYp**, **sJcA**), realistic (**YK4X**), and well-motivated (**sJcA**) solution for reducing labeling cost (**6wYp**, **YK4X**, **QtBh**). We also appreciate their positive comments on the use of conformal prediction in active learning (**6wYp**, **YK4X**), the strong empirical performance of our method (**6wYp**, **sJcA**), and the thorough ablation studies (**6wYp**, **sJcA**). They also recognized that the paper is well written (**YK4X**, **sJcA**) and theoretically sound (**sJcA**).

### **Summary of Rebuttal / Discussions**

We summarized the key comments and our corresponding responses below:

| Comment | Response |
| :--- | :--- |
| Clarification of our acquisition function (**6wYp**) and empirical quantile (**QtBh**) | We have added detailed explanations of the acquisition function and empirical quantile, and updated the manuscript accordingly. |
| Guidelines for setting hyperparameter $d$ (**6wYp**, **sJcA**) | We have conducted ablation studies on the hyperparameter $d$ on CIFAR-10, CIFAR-100, and ImageNet64x64, and provided a guideline for setting $d$ based on the observations. Reviewer **sJcA** expressed satisfaction with our responses. |
| Effectiveness of our method on uncertain samples (**6wYp**, **YK4X**) | We have shown that our method outperforms the sift-out baseline suggested by reviewer **6wYp** when using entropy sampling that favors uncertain samples. We have further justified that our method is compatible with diverse uncertainty measures. |
| Dependency on calibration set and quantile from previous round (**YK4X**) | We have clarified that the calibration set is also used for model training. We have also provided results showing that our method is robust to the size of the calibration set and remains highly effective without using the quantile from the previous round. |
| Realistic aspect of our cost model (**YK4X**) | We have explained the motivation behind our cost model, supported by the user study involving 40 participants. |
| Theoretical analysis on label complexity and its convergence (**YK4X**) | We have clarified that candidate set query is orthogonal to the acquisition function, ensuring compatibility with acquisition functions with label complexity analysis. We have also discussed the challenges of proving the label complexity under deep learning models. |
| Experiment on more realistic datasets (**sJcA**, **QtBh**) | We have conducted experiments on datasets with noisy labels and class imbalances and on a text classification dataset, demonstrating robustness on various realistic scenarios. Reviewer **sJcA** acknowledged the generalizability of our method across modalities and its robustness to real-world datasets. |
| Experiment on additional acquisition function (**sJcA**) | We have evaluated our method with the ProbCover (Yehuda et al., 2022) acquisition function, confirming its compatibility with advanced acquisition methods. |


### **Before/After Rebuttal**

- Reviewer **6wYp**: We believe our extensive experiments on the requested scenarios, along with the provided explanations, have addressed most of the concerns.
- Reviewer **YK4X**: The reviewer noted that several concerns, including confidence level settings, the cost model, and calibration set sampling, had been addressed but noted that a few concerns remained. To resolve these, we have provided additional experiments and clarifications.
- Reviewer **sJcA**: The reviewer stated that the responses satisfy most of their concerns and that they would **rate to accept** this paper.
- Reviewer **QtBh**: We believe our clarifications and extensive additional experiments have addressed the concerns raised by the reviewer, who was already positive with our work with a score of 6.


---
---

### **References**

Yehuda, O., Dekel, A., Hacohen, G., & Weinshall, D. (2022). Active learning through a covering lens. Advances in Neural Information Processing Systems, 35, 22354–22367.

---

### Meta-Review · Area_Chair_MfFm · 2024-12-23

**Metareview:**

This paper was reviewed by four experts in the field and received 5, 5, 6, 6 as the final ratings. The reviewers agreed that the paper proposes an interesting active learning (AL) technique to further reduce the labeling burden on the human annotators, the proposed solution utilizes the well-established theory of conformal predictions, the experimental results are encouraging, and that the content of the paper is well-presented.


The authors have mentioned that their candidate set query design is independent of the sampling method, and it can be applied in conjunction with any acquisition function. However, in their paper, the authors have demonstrated their method with only three AL acquisition functions: random sampling, entropy sampling and BADGE sampling. Given the plethora of AL sampling functions (such as those based on uncertainty, diversity, representativeness and their combinations), demonstrating the performance with only three functions (out of which random and entropy are very naïve methods) is not sufficient. While the authors have conducted experiments in the rebuttal using the ProbCover acquisition function, in response to Reviewer sJcA's comment, the AC feels that a more thorough investigation of the proposed candidate set query method with more advanced acquisition functions is necessary to appropriately understand its merit and usefulness.


The active learning setting proposed in this paper narrows down the set of candidate classes likely to contain the ground truth class, so that the oracle won’t have to examine all possible classes. This setting is very similar to that proposed in the following papers:

[1] Joshi et al. "Breaking the interactive bottleneck in multi-class classification with active selection and binary feedback". CVPR 2010

[2] Bhattacharya and Chakraborty. "Active Learning with n-ary Queries for Image Recognition". WACV 2019

These methods should be used as comparison baselines; it is difficult to assess the merit of the proposed method without a thorough comparison against related techniques.


A concern was also raised about the convergence guarantee of the proposed acquisition function. While the authors have acknowledged this in the rebuttal, and have mentioned that this is due to the reliance on heuristic informativeness measures, establishing convergence guarantees will further strengthen the theoretical aspects of the proposed solution. The authors are encouraged to look into this aspect.


We appreciate the authors' efforts in meticulously responding to each reviewer’s comments. We also greatly appreciate their efforts in conducting additional experiments to address some of the reviewers' concerns (such as the experiments on datasets containing imbalance classes, experiments on datasets containing label noise, experiments with the ProbCover acquisition function and experiments on text classification). However, in light of the above discussions, we conclude that the paper may not be ready for an ICLR publication in its current form. While the paper clearly has merit, the decision is not to recommend acceptance. The authors are encouraged to consider the reviewers' comments when revising the paper for submission elsewhere.

**Additional Comments On Reviewer Discussion:**

Please see my comments above.

---

### Decision · Program_Chairs · 2025-01-22

Reject